# PROMISE: PROMPT-ROBUST VISION-LANGUAGE MODELS VIA META-FINETUNING

**Haohui Liang**[1*]**, Runlin Huang**[1*]**, Yingjun Du**[2]**, Yujia Hu**[1]**, Weifeng Su**[1]**, Cees G. M. Snoek**[2†]

[1] Beijing Normal-Hong Kong Baptist University
[2] University of Amsterdam
[*] Equal contribution.
[†] Corresponding author.

## ABSTRACT

Vision-language models (VLMs) have demonstrated remarkable generalization across diverse tasks by leveraging large-scale image-text pretraining. However, their performance is notoriously unstable under variations in natural language prompts, posing a considerable challenge for reliable real-world deployment. To address this prompt sensitivity, we propose **Promise**, a meta-learning framework for **prom**pt-robust **vi**sion-languag**e** models via meta-finetuning, which explicitly learns to generalize across diverse prompt formulations. Our method operates in a dual-loop meta-finetuning setting: the inner loop adapts token embeddings based on a set of varied prompts, while the outer loop optimizes for generalization on unseen prompt variants. To further improve robustness, we introduce an adaptive prompt weighting mechanism that dynamically emphasizes more generalizable prompts and a token-specific learning rate module that fine-tunes individual prompt tokens based on contextual importance. We further establish that **Promise**'s weighted and preconditioned inner update provably (i) yields a one-step decrease of the outer empirical risk together with a contraction of across-prompt sensitivity, and (ii) tightens a data-dependent generalization bound evaluated at the post-inner initialization. Across 15 benchmarks spanning base-to-novel generalization, cross-dataset transfer, and domain shift, our approach consistently reduces prompt sensitivity and improves performance stability over existing prompt learning methods.

## 1 INTRODUCTION

Vision-language models (VLMs) have achieved impressive generalization by aligning image and text representations through large-scale pretraining (Radford et al., 2021; Jia et al., 2021; Li et al., 2022). A common zero-shot inference strategy in these models involves filling handcrafted templates like "`a photo of a [CLASS]`" and comparing encoded text features against image embeddings. However, their predictions are surprisingly brittle: small variations in prompt phrasing can cause large performance fluctuations (Zhou et al., 2022b). This phenomenon, often referred to as *prompt sensitivity*, presents a serious obstacle when deploying VLMs reliably in real-world applications.

To mitigate manual prompt engineering, recent efforts in prompt learning introduce learnable tokens to replace or augment natural language prompts (Lester et al., 2021; Zhou et al., 2022a; Jia et al., 2022). CoOp (Zhou et al., 2022b) and CoCoOp (Zhou et al., 2022a) optimize textual prompts with supervision from few-shot examples but tend to overfit on base classes and generalize poorly to novel ones. Khattak et al. (2023b) propose MaPLe, a multi-modal approach that injects visual and textual prompts across network layers to improve transferability. More recently, Guo & Gu (2025) enhance cross-modal representations via dynamic feature routing. While these approaches improve downstream adaptation, they still rely on a fixed set of prompt templates during finetuning and inference. As shown in Figure 1, they remain vulnerable to prompt rewording and offer limited robustness to natural linguistic variability.

This paper addresses the underexplored challenge of building *prompt-robust* vision-language models, models whose predictions remain stable across diverse prompt formulations. While prior meta-learning-based prompt tuning methods (Li et al., 2023; Park et al., 2024; Zhao et al., 2024) have

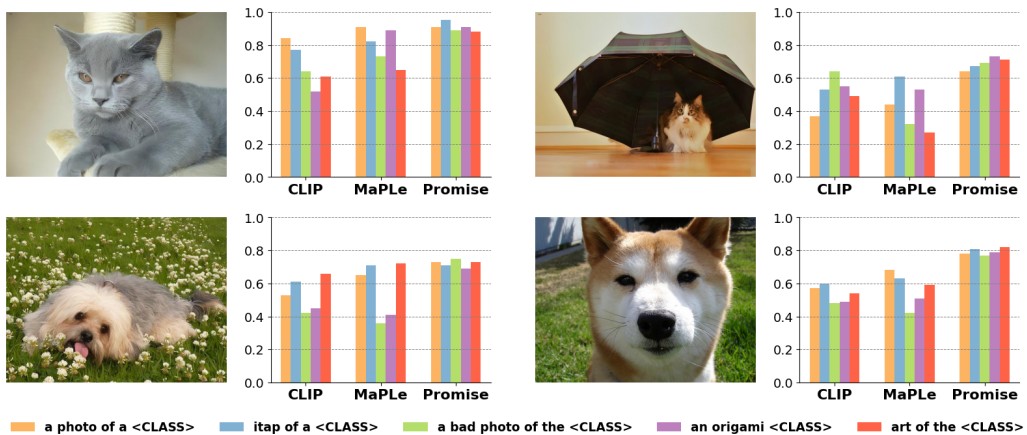

**Figure 1: Prompt sensitivity and performance comparison in VLMs.** We visualize the predicted confidence scores across five prompt formulations for the same input image. CLIP and MaPLe exhibit high sensitivity to prompt phrasing, with both larger fluctuations and lower overall scores. In contrast, our method not only produces more consistent predictions across prompts, but also achieves higher confidence values, reflecting both improved robustness and better overall performance.

improved generalization by learning better prompt initializations, domain-invariant representations, or task-level regularization, they typically assume fixed prompt templates and focus on cross-task or cross-domain transfer. In contrast, we target *intra-task robustness* by explicitly modeling variation across semantically equivalent prompts within the same task. Our method addresses this gap through a dual-loop meta-learning framework that learns prompt-invariant token embeddings and dynamically adapts to prompt-level variability.

To address prompt sensitivity, we introduce **Promise**, a meta-learning framework for **prom**pt-robust v**is**ion-languag**e** models via meta-finetuning. The model learns to adapt token embeddings under varying prompt views and generalizes to unseen phrasings. Inspired by prior work on meta-gradient adaptation, we build a framework that enables both prompt-level and token-level adaptation. Our approach makes four key contributions. First, we design a dual-loop finetuning strategy: the inner loop adapts learnable token embeddings using a subset of prompt templates, while the outer loop evaluates them on disjoint prompts to encourage prompt-invariant representations. Second, we incorporate an adaptive prompt weighting mechanism that dynamically prioritizes templates with higher generalization utility. Third, we develop a token-specific learning rate module, which fine-tunes each prompt token based on its contextual importance. These components work jointly to produce stable, generalizable predictions across prompt variations. Finally, we prove that **Promise** ' weighted, preconditioned inner step both decreases the outer empirical risk in one move and contracts across-prompt sensitivity, thereby tightening a data-dependent generalization bound at the post-inner initialization.

Extensive experiments across 15 datasets demonstrate that our framework substantially reduces prompt sensitivity while enhancing generalization to novel categories. It consistently outperforms prior multi-modal prompt learning methods on base-to-new transfer, cross-dataset adaptation, and domain generalization benchmarks. Ablation studies further validate the effectiveness of each component, confirming that meta-learning, adaptive prompt weighting, and token-specific learning rates jointly contribute to improved robustness and reduced sensitivity to prompt variation in vision-language models.

## 2 RELATED WORK

**Learning the Prompt Template.** This line of work treats the textual prompt as a learnable sentence or soft vector sequence. CoOp (Zhou et al., 2022b) and CoCoOp (Zhou et al., 2022a) pioneered this approach by optimizing continuous embeddings of the prompt template in the language branch of CLIP (Radford et al., 2021), enabling few-shot generalization to new classes. Variants such as Bayesian prompt learning (Derakhshani et al., 2023) further model uncertainty over soft prompts to improve generalization. Subsequent works explore different strategies to enhance prompt robust-

ness and generalizability: GRAM (Li et al., 2023) and MetaPrompt (Zhao et al., 2024) introduce meta-learning-based schemes to improve adaptation across domains, while DePT (Zhang et al., 2024) alleviates overfitting by decoupling base-specific and task-shared knowledge during prompt optimization. These methods directly modify the textual input space, typically focusing on adapting the entire prompt string. In contrast, our work targets a different axis of generalization—robustness to prompt phrasing variation within the same task. Rather than optimizing a fixed prompt template, we simulate prompt distribution shifts during training and meta-learn prompt-invariant token embeddings. This shift in focus—from task-level or domain-level generalization to within-task prompt robustness—is central to our design and has been underexplored in prior prompt tuning literature.

**Learning the Prompt Tokens.** An alternative and increasingly prevalent direction is to append learnable tokens to the input embeddings without altering the prompt text itself. This paradigm, often referred to as prefix tuning, has been applied to either the text encoder (Khattak et al., 2025), the vision encoder (Jia et al., 2022; Bahng et al., 2022), or both (Li et al., 2024; Khattak et al., 2023b; Roy & Etemad, 2024; Guo & Gu, 2025; Park et al., 2024). These methods introduce soft tokens across one or more layers to improve model adaptation without modifying the underlying language prompt. Despite their effectiveness, they still rely on fixed prompt templates during training and evaluation, making them vulnerable to prompt phrasing variations. Our work belongs to this category but goes beyond conventional prompt tokens by incorporating meta-learning to explicitly learn from prompt variations. Instead of optimizing prompt tokens for a single fixed formulation, our framework learns to generalize across diverse prompt templates, thereby improving robustness to natural language variation.

## 3 METHOD

We propose prompt-robust vision-language models via Meta-finetuning (**Promise**), a meta-learning framework that enhances the robustness of vision-language models (VLMs) to variations in prompt phrasing. Our method learns prompt-invariant token representations by simulating prompt distribution shifts during finetuning. Specifically, it adopts a dual-loop structure: the inner loop updates token embeddings using a subset of prompt templates, while the outer loop optimizes for generalization to unseen prompts. Additionally, our method introduces two key modules—(1) an adaptive prompt weighting mechanism to prioritize generalizable prompts, and (2) a token-specific learning rate scheduler for fine-grained token adaptation. These components jointly improve stability and performance under prompt variation.

### 3.1 DUAL-LOOP META-FINETUNING

Let $\mathcal{P}_{\text{in}}$ and $\mathcal{P}_{\text{out}}$ denote the subsets of prompt templates used in the inner and outer loops, respectively, where $\mathcal{P}_{\text{in}} \cap \mathcal{P}_{\text{out}} = \emptyset$. For each prompt $\mathbf{T}_i \in \mathcal{P}_{\text{in}}$, we associate a learnable token $\boldsymbol{e}_i$. The meta-parameters $\theta$ represent the initial shared parameters for all prompt-specific tokens, where $\theta_t$ denotes the parameters associated with text tokens, and $\theta_v$ denotes those associated with visual features. For simplicity of notation, we refer to these collectively as $\theta$ from now on. In our instantiation, the "tasks" in the meta-learning sense correspond to different natural-language prompt variants within the same dataset, rather than to different datasets or label spaces. The dual-loop structure is therefore used to explicitly simulate prompt distribution shifts within a fixed label space.

**Inner-Loop Finetuning.** In the inner loop, we adapt the text token embeddings $\boldsymbol{e}_i$ for each prompt template $\mathbf{T}_i \in \mathcal{P}_{\text{in}}$. Given an input image $\mathbf{x}$ and its corresponding label $y$, the model adapts $\boldsymbol{e}_i$ to minimize the loss for each prompt in $\mathcal{P}_{\text{in}}$. This process can be described as follows:

$$\hat{\theta} = \theta - \alpha \sum_{\mathbf{T}_i \in \mathcal{P}_{\text{in}}} \nabla_\theta \mathcal{L}_{\text{in}}(\mathbf{f}(\boldsymbol{e}_i, \mathbf{x}), y), \tag{1}$$

where $\mathcal{L}_{\text{in}}(\cdot)$ represents the loss computed with prompts from $\mathcal{P}_{\text{in}}$, $\alpha$ is the learning rate, and $\mathbf{f}(\boldsymbol{e}_i, \mathbf{x})$ denotes the model's output given the adapted prompt embedding $\boldsymbol{e}_i$ and input $\mathbf{x}$. This prompt-specific adaptation helps the model capture the variations within the prompts in $\mathcal{P}_{\text{in}}$.

**Outer-Loop Finetuning.** In the outer loop, we optimize the meta-parameters $\theta$ with a distinct set of prompt templates $\mathcal{P}_{\text{out}}$. For each prompt template $\mathbf{T}_j \in \mathcal{P}_{\text{out}}$, the corresponding token embedding $\boldsymbol{e}_j$

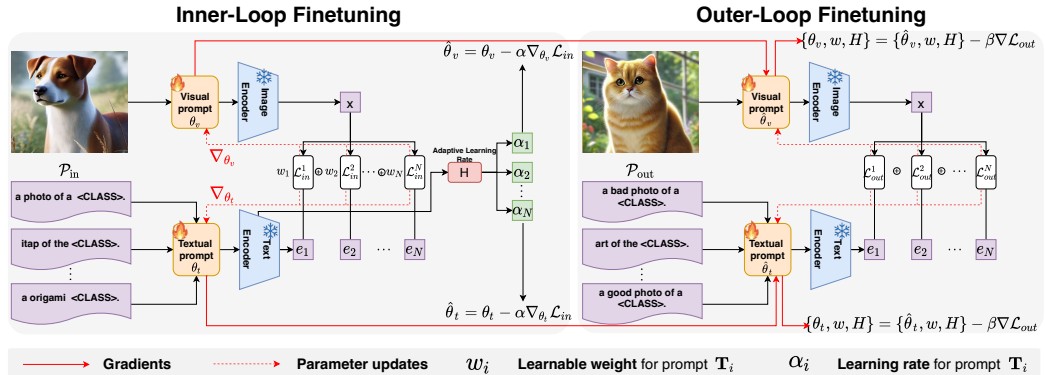

**Figure 2: Overview of Promise.** Inner-loop finetuning adapts the prompt tokens using the subset of prompts in $\mathcal{P}_{\text{in}}$ to minimize the loss. Outer-loop finetuning updates the meta-parameters $\theta$ across the disjoint prompt subset $\mathcal{P}_{\text{out}}$ to promote prompt-agnostic generalization. Adaptive prompt weighting and token-specific learning rates further stabilize performance under prompt variation.

is utilized. Leveraging the adapted parameters $\hat{\theta}$ from the inner loop, we refine $\theta$ by minimizing the aggregate loss over all prompts in $\mathcal{P}_{\text{out}}$:

$$\theta = \theta - \beta \sum_{\mathbf{T}_j \in \mathcal{P}_{\text{out}}} \nabla_{\hat{\theta}} \mathcal{L}_{\text{out}}(\mathbf{f}(\boldsymbol{e}_j, \mathbf{x}), y), \tag{2}$$

where $\mathcal{L}_{\text{out}}(\cdot)$ is the loss calculated with prompts from $\mathcal{P}_{\text{out}}$ and $\beta$ is the outer-loop learning rate. This process ensures that the meta-parameters $\theta$ are optimized to generalize across both $\mathcal{P}_{\text{in}}$ and $\mathcal{P}_{\text{out}}$, achieving the intended prompt-agnostic performance. In practice, this inner/outer separation allows the model to adapt on one subset of prompt templates and to be explicitly optimized for stable performance on disjoint, unseen phrasings of the same labels, which directly targets intra-task prompt robustness.

While MAML (Finn et al., 2017) provides a general framework for meta-gradient adaptation through an inner- and outer-loop structure, it typically assumes uniform treatment of input samples during meta-training. Our formulation is MAML-inspired in that it uses a similar dual-loop meta-finetuning structure, but it is instantiated specifically for prompt robustness in vision–language models by treating different prompt templates as tasks and by explicitly optimizing for consistency across $\mathcal{P}_{\text{in}}$ and $\mathcal{P}_{\text{out}}$. Inspired by this dual-loop formulation, we adopt a similar meta-learning structure to simulate prompt variation: the inner loop performs prompt-specific adaptation, while the outer loop promotes generalization across prompt distributions. However, unlike MAML, it applies a uniform update across all inputs and lacks the flexibility to differentiate between prompts of varying generalization quality. This limitation is particularly problematic in prompt tuning, where some prompt templates are inherently more transferable than others.

## 3.2 ADAPTIVE PROMPT WEIGHTING

To further enhance robustness across varied prompt templates, **Promise** incorporates an adaptive prompt weighting mechanism. In this approach, each text prompt template $\mathbf{T}_i$ is assigned a learnable weight $w_i$ within the inner loop, which scales the loss corresponding to each prompt template. This allows the model to learn the importance of each template dynamically, similar to mixture of expert models. The weights $\{w_i\}_{i=1}^{N}$ are optimized in the outer loop, where $N$ represents the number of prompts in the inner loop. This optimization enables the model to prioritize templates that contribute more effectively to generalization.

In the inner loop, for each prompt template $\mathbf{T}_i \in \mathcal{P}_{\text{in}}$, we compute the prompt-specific loss $\mathcal{L}_{\text{in}}^i$ weighted by the corresponding parameter $w_i$. The adapted parameter $\theta_i$ for each prompt $\mathbf{T}_i$ is updated as:

$$\hat{\theta} = \theta - \alpha \sum_{\mathbf{T}_i \in \mathcal{P}_{\text{in}}} w_i \, \nabla_{\theta} \mathcal{L}_{\text{in}}(\mathbf{f}(\boldsymbol{e}_i, \mathbf{x}), y), \tag{3}$$

where $\mathcal{L}_{\text{in}}(\cdot)$ represents the inner-loop loss for prompt $\mathbf{T}_i$, $\alpha$ is the learning rate, and $w_i$ modulates the impact of each prompt-specific loss in the inner loop. This weighted adaptation enables the model to learn varying levels of emphasis for different prompt templates during the inner-loop optimization.

After the inner-loop updates, the outer loop optimizes the meta-parameters $\theta$ together with a set of unnormalized prompt scores $\{s_i\}_{i=1}^N$ on $\mathcal{P}_{\text{out}}$. To aggregate across prompts while controlling variance, we constrain the resulting weights $\tilde{\mathbf{w}} = (\tilde{w}_1, \ldots, \tilde{w}_N)$ to lie on the probability simplex $\Delta^{N-1} = \{\mathbf{w} \geq 0, \sum_i w_i = 1\}$ via an entropic mirror map with a temperature $\tau > 0$: the outer step performs a gradient update on $\{s_i\}$ followed by a mirror descent normalization onto $\Delta^{N-1}$. The temperature $\tau$ (learned or annealed) trades off exploration and concentration: larger $\tau$ spreads mass across templates to stabilize gradients, while smaller $\tau$ concentrates mass on high-utility prompts to reduce estimator variance. To further prevent degenerate collapse and encourage useful sparsity, we include an optional entropy regularizer $\lambda \mathcal{H}(\tilde{\mathbf{w}})$ (or its sparse alternative via entmax), which keeps a few high-utility prompts active while pruning noisy ones. Intuitively, this simplex-constrained, temperature-controlled weighting focuses learning signal on prompts that best transfer to unseen phrasings, without hand-tuning per-template coefficients.

Finally, the outer-loop update for both the meta-parameters $\theta$ and the weights $\{w_i\}$ is defined as: $\theta = \theta - \beta \nabla_{\hat{\theta}} \mathcal{L}_{\text{out}}$, $w_i = w_i - \beta \nabla_{w_i} \mathcal{L}_{\text{out}}$ where $\beta$ is the learning rate for the outer-loop optimization. This update process enables the model to adaptively assign higher importance to prompts that contribute more effectively to generalization, achieving a more robust prompt-agnostic performance.

### 3.3 Token-Specific Adaptive Learning Rate

In **Promise**, we implement a token-specific adaptive learning rate module using a neural network that generates learning rates for each token embedding in the inner loop. Our design is inspired by MetaSGD (Li et al., 2017), which learns a task-specific learning rate for each parameter. We extend this idea by introducing a token-wise, data-driven learning rate module. Instead of treating learning rates as fixed or per-task scalars, we use a neural network to generate prompt-token-specific learning rates conditioned on input features. This enables fine-grained, context-aware adaptation for each token within the inner loop. Specifically, given a set of prompt templates $\mathbf{T}_i$ and their corresponding features extracted by the text encoder, $H(\cdot)$ generates a unique learning rate $\alpha_i$ for each learnable token $\boldsymbol{e}_i$ in prompt template $\mathbf{T}_i$.

In the inner loop, the token-specific learning rate $\alpha_i$ is used to update each token embedding $\boldsymbol{e}_i$ in a targeted way. The adaptation process for each token embedding $\boldsymbol{e}_i$ with the generated learning rate is as follows:

$$\hat{\theta} = \theta - \alpha_i \sum_{\mathbf{T}_i \in \mathcal{P}_{\text{in}}} w_i \nabla_\theta \mathcal{L}_{\text{in}}(\mathbf{f}(\boldsymbol{e}_i, \mathbf{x}), y), \tag{4}$$

where $\mathcal{L}_{\text{in}}(\cdot)$ represents the inner-loop loss for prompt template $\mathbf{T}_i$, and $\alpha_i = H(\boldsymbol{e}_i)$ is the data-driven, token-specific learning rate generated by $H(\cdot)$. This token-specific adjustment enables the model to fine-tune each token based on its importance in the prompt, improving adaptation to variations in prompt templates. $H(\cdot)$ takes the encoded features of each prompt template $\mathbf{T}_i$ as input and outputs a unique learning rate $\alpha_i$ for each token embedding $\boldsymbol{e}_i$. This data-driven approach allows the learning rates to be conditioned on the specific characteristics of each prompt template, rather than using a uniform rate across all tokens. Formally, the token-specific learning rate for each token $\boldsymbol{e}_i$ is defined as $\alpha_i = H(\boldsymbol{e}_i)$, where $\boldsymbol{e}_i$ denotes the encoded features of prompt $\mathbf{T}_i$ derived from the text encoder. This feature-driven learning rate allows the model to adapt each token's learning rate based on the semantic and contextual information within the prompt template.

In the outer loop, we update both the meta-parameters $\theta$ and the parameters of $H(\cdot)$ to optimize the model's generalization across different prompt templates. The outer-loop objective thus becomes: $\theta = \theta - \beta \nabla_{\hat{\theta}} \mathcal{L}_{\text{out}}$, $H = H - \beta \nabla_H \mathcal{L}_{\text{out}}$, where $\mathcal{L}_{\text{out}}$ is the cumulative outer-loop loss across prompts in $\mathcal{P}_{\text{out}}$ and $\beta$ is the outer-loop learning rate. This process ensures that both the meta-parameters and $H(\cdot)$ are optimized to enhance prompt-agnostic performance across diverse prompt templates.

## 4 ANALYSIS

The core idea of our **Promise** is to turn a single, globally shared SGD initialization into a prompt-aware initialization via a weighted and preconditioned inner step. The outer optimization starts from a better-conditioned and lower-variance point by tailoring the inner update to the prompt distribution—through adaptive prompt weights and a token-wise diagonal preconditioner. In this section, we theoretically analyze the advantage of this adaptation: we show (i) a guaranteed one-step decrease of the outer empirical risk together with a contraction of across-prompt sensitivity, and (ii) a tightened, data-dependent generalization bound evaluated at the post-inner initialization.

**Setup.** For a prompt template $T$, let $\mathcal{D}_T$ denote its data distribution over samples $z = (x, y)$, and let $\ell(\theta; z, T)$ be the per-sample loss. We define the *population outer risk*

$$R_{\text{pop}}(\theta) := \mathbb{E}_{T \sim \mathcal{P}} \, \mathbb{E}_{z \sim \mathcal{D}_T} \left[ \ell(\theta; z, T) \right].$$

Given a finite set of samples $\{z_{T,j}\}_{j=1}^{n_T}$ for each $T$, the per-prompt empirical loss is

$$L(\theta; T) = \frac{1}{n_T} \sum_{j=1}^{n_T} \ell(\theta; z_{T,j}, T), \quad \widehat{R}_{P_{\text{out}}}(\theta) = \frac{1}{|P_{\text{out}}|} \sum_{T \in P_{\text{out}}} L(\theta; T), \quad n_{\text{out}} := \sum_{T \in P_{\text{out}}} n_T.$$

In each meta-episode we draw *independent* and disjoint prompt subsets $(P_{\text{in}}, P_{\text{out}})$ with $P_{\text{in}} \cap P_{\text{out}} = \emptyset$. **Promise** performs a *weighted & preconditioned* inner update

$$\hat{\theta} = \theta - P \sum_{T \in P_{\text{in}}} w_T \, \nabla_\theta L(\theta; T), \qquad P = \text{diag}(\alpha_1, \dots, \alpha_d) \succeq 0, \quad \sum_T w_T = 1, \; w_T \geq 0,$$

and then evaluates $\widehat{R}_{P_{\text{out}}}$ at $\hat{\theta}$ to update $(\theta, w, H)$. For later use, denote the aggregated inner gradient and its variance by

$$G(\theta) := \sum_{T \in P_{\text{in}}} w_T \, \nabla_\theta L(\theta; T), \quad \mu(\theta) := \mathbb{E}[G(\theta)], \quad \Sigma_w := \text{Var}[G(\theta)].$$

**Assumptions.** (i) $L(\cdot; T)$ is $L$-smooth; (ii) stochastic gradients are unbiased with bounded second moment; (iii) $P_{\text{in}}$ and $P_{\text{out}}$ are independent draws from $\mathcal{P}$; (iv) the weighted inner gradient aligns with the outer descent direction (Def. B.1).

**Theorem 4.1** (One-step outer-risk descent & sensitivity contraction). *Under the assumptions above, there exist constants $\gamma \in (0, 1]$, $\kappa \geq 1$ determined by the alignment of the weighted inner gradient and by gradient heterogeneity across prompts, such that*

$$\mathbb{E}\left[ \widehat{R}_{P_{\text{out}}}(\hat{\theta}) \right] \leq \widehat{R}_{P_{\text{out}}}(\theta) - \left( \gamma \, \lambda_{\min}(P) - \frac{L}{2} \kappa^2 \|P\|_2^2 \right) \left\| \nabla \widehat{R}_{P_{\text{out}}}(\theta) \right\|^2 + \frac{L}{2} \|P\|_2^2 \, \text{Tr}(\Sigma_w),$$

*where $\Sigma_w := \text{Var}\left[ \sum_{T \in P_{\text{in}}} w_T \, \nabla_\theta L(\theta; T) \right]$. In particular, if $\|P\|_2$ is small enough so that $\gamma \, \lambda_{\min}(P) > \frac{L}{2} \kappa^2 \|P\|_2^2$, then the expected outer risk strictly decreases. Moreover, defining the across-prompt sensitivity $S(\theta) := \text{Var}_{T \sim \mathcal{P}}[f_\theta(x; T)]$ (or its Jacobian surrogate), we have*

$$\mathbb{E}\left[ S(\hat{\theta}) \right] \leq (1 - \mu_{\text{eff}}) S(\theta) + C \, \|P\|_2^2 \, \text{Tr}(\Sigma_w),$$

*for some $\mu_{\text{eff}} > 0$ depending on the local Lipschitz constants of $f_\theta$ and the same alignment, i.e., the inner step contracts prompt sensitivity up to a variance term that shrinks with $\text{Tr}(\Sigma_w)$.*

The adaptive weights $w_T$ concentrate the inner update on prompts whose gradients are *aligned* with the outer descent direction (variance reduction), while the token-wise diagonal $P$ rescales ill-conditioned coordinates (preconditioning). Their combination enlarges the one-step decrease of the outer objective and suppresses across-prompt drift. Intuitively, the resulting inner update

$$\theta^+ = \theta - P \, \nabla_\theta \mathcal{L}_w(\theta), \quad \mathcal{L}_w(\theta) := \sum_{T \in P_{\text{in}}} w_T \, L(\theta; T),$$

acts as a cautious, biased gradient step that moves more strongly along directions supported by prompts which generalize well to $P_{\text{out}}$, while dampening directions that are unstable across templates.

Under the conditions of Theorem 4.1, this weighted and preconditioned step tends to shrink the prompt-averaged outer risk, which is what we refer to as a "risk contraction" effect.

Empirically, we validate this behavior by counting how often a single inner update decreases the outer empirical loss on $P_{\text{out}}$. On a representative subset of datasets, the fraction of such "risk-decreasing" inner steps increases from about 61% for an unweighted, unpreconditioned update to about 78% when using the full **Promise** update with both $w$ and $P$. This matches the qualitative picture suggested by Theorem 4.1, even though the smoothness and alignment assumptions used in the analysis are only approximate for large CLIP-scale networks in practice.

**Theorem 4.2** (Data-dependent generalization at the post-inner initialization). *Let* $\hat{\theta} = \theta - P \sum_{T \in P_{\text{in}}} w_T \nabla_\theta L(\theta; T)$. *Conditioned on* $\hat{\theta}$, *the samples in* $P_{\text{out}}$ *are i.i.d.; hence for any* $\delta \in (0, 1)$, *with probability at least* $1 - \delta$,

$$R_{\text{pop}}(\hat{\theta}) \ \leq \ \widehat{R}_{P_{\text{out}}}(\hat{\theta}) \ + \ c_2 \sqrt{\frac{\ln(1/\delta)}{n_{\text{out}}}} \ + \ c_1 \, \Gamma \, \sqrt{\text{Tr}(P \, \Sigma_w \, P)}.$$

*Here* $n_{\text{out}}$ *is the number of samples used by* $\widehat{R}_{P_{\text{out}}}$, $\Gamma$ *upper-bounds the* $\theta$-*Lipschitz constant of* $L(\theta; T)$, *and the last term quantifies the initialization variability induced by the inner-step gradient noise. Consequently, any choice of* $(w_T, P)$ *that reduces both* $\widehat{R}_{P_{\text{out}}}(\hat{\theta})$ *and* $\text{Tr}(P\Sigma_w P)$ *tightens the bound on* $R_{\text{pop}}(\hat{\theta})$.

The complete proof is provided in Appendix B.

## 5 EXPERIMENTS

### 5.1 EXPERIMENTAL SETUP

**15 Datasets.** To evaluate base-to-new generalization and cross-dataset generalization, we adopt a diverse set of 11 image classification datasets, following prior work such as CLIP (Radford et al., 2021) and CoOp (Zhou et al., 2022b). These datasets cover a wide range of visual recognition tasks: ImageNet (Deng et al., 2009) and Caltech101 (Fei-Fei et al., 2004) are used for generic object classification; OxfordPets (Parkhi et al., 2012), StanfordCars (Krause et al., 2013), Flowers102 (Nilsback & Zisserman, 2008), Food101 (Bossard et al., 2014), and FGVCAircraft (Maji et al., 2013) are included for fine-grained image recognition; EuroSAT (Helber et al., 2019) is employed for satellite image classification; UCF101 (Soomro et al., 2012) for action recognition; DTD (Cimpoi et al., 2014) for texture classification; and SUN397 (Xiao et al., 2010) for scene recognition. For domain generalization experiments, we follow CoOp (Zhou et al., 2022b) and use ImageNet as the source domain, with four distinct ImageNet variants serving as the target domains: ImageNetV2 (Recht et al., 2019), ImageNet-Sketch (Wang et al., 2019), ImageNet-A (Hendrycks et al., 2021b), and ImageNet-R (Hendrycks et al., 2021a).

**Implementation Details.** To ensure a fair comparison, we use the CLIP-ViT-B/16 architecture as the base model across all methods, consistent with previous work like CoCoOp (Zhou et al., 2022a) and VPT (Jia et al., 2022). In line with MaPLe (Khattak et al., 2023a), we set the prompt depth to 9 and used prompt lengths of 2 for both language and vision prompts. All models are trained for 10 epochs with a batch size of 8 and a learning rate of 0.0035, utilizing the SGD optimizer on a single NVIDIA A6000 GPU. For training on the full 1000 classes of ImageNet as the source model, we set the prompt depth to 3 and trained for 5 epochs with a learning rate of 0.0035. Following PromptSRC (Khattak et al., 2023b), we adopt the 60 hand-crafted prompt templates originally provided in the PromptSRC appendix (Khattak et al., 2023b). In each training episode, we randomly sample 30 templates for inner-loop adaptation and 30 disjoint templates for outer-loop generalization. Our network $H$ consists of a 3-layer MLP. We perform three iterations for the inner loop and adopt Reptile (Nichol et al., 2018)'s, using first-order derivatives to approximate the outer-loop loss. For consistency, all the results of the learning-based methods are computed as an average over three random seeds. All code will be made available. Additional results are in the Appendix: more robustness visualizations (App. §C), training time analysis (App. §G.2), results of structured-prompt effects (App. §D), detailed full prompts (App. §E), and algorithmic details (App. §F).

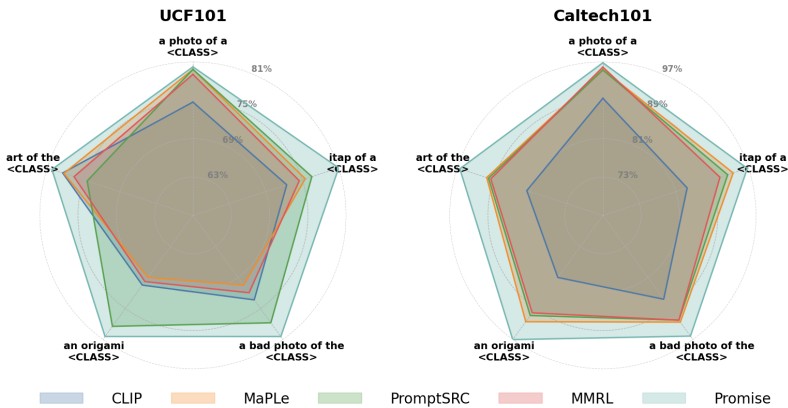

Figure 3: **Sensitivity of models to various prompts** on UCF101 and Caltech101. While other methods show considerable variability across templates, our method maintains consistent high performance, demonstrating robustness to prompt variation.

## 5.2 RESULTS

**Prompt Sensitivity of Promise.** To assess robustness to prompt variation, we use MaPLe as the base prompt learner and apply our method on top, resulting in the combined model **Promise**, we compare CLIP, MaPLe, PromptSRC, and MMRL under diverse prompt templates. This analysis highlights the ability to maintain both high accuracy and stable predictions across formulations, achieving prompt-agnostic behavior. We begin with a quantitative study on the UCF101 and Caltech101 datasets, evaluating the consistency of predictions across different templates. As shown in Figure 3, while other methods exhibit notable performance fluctuations across prompts, our method yields consistently high accuracy regardless of template choice. This stability underscores the effectiveness of our approach in mitigating prompt sensitivity and ensuring reliable performance under natural prompt shifts.

**Enhancing Existing Prompt Learners.** Table 1 shows that our meta-learning framework consistently improves the performance of existing prompt learning methods across 11 datasets. We apply our training strategy on top of MaPLe, IVLP, and MMRL by keeping their base architectures unchanged while replacing their original optimization with our dual-loop meta-learning procedure. Notably, this yields considerable gains in novel class accuracy—e.g., +2.09 on IVLP, +2.12 on MaPLe, and +1.68 on MMRL—while maintaining or slightly improving base class performance. These gains result in higher harmonic means across all methods, demonstrating that our framework not only enhances generalization but also integrates well with diverse prompting strategies.

Table 1: **Effect of Promise**.

| Method | Base | New | H |
|---|---|---|---|
| IVLP | 82.51 | 73.35 | 77.66 |
| + Promise | 83.26 | 75.44 | 79.16 |
| MaPLe | 82.28 | 75.14 | 78.55 |
| + Promise | 83.57 | 77.26 | 80.29 |
| MMRL | 85.68 | 77.16 | 81.20 |
| + Promise | 86.14 | 78.84 | 82.33 |

**Benefit of Adaptive Prompt Weighting.** We assess the impact of adaptive prompt weighting by comparing **Promise** variants with and without this component, using MaPLe and MMRL as backbones. As shown in Table 2, incorporating adaptive weighting consistently improves performance, particularly on novel classes and in harmonic mean, across both settings. This mechanism complements our meta-finetuning framework by enabling the model to dynamically prioritize more transferable prompts during adaptation, thereby enhancing optimization efficiency and generalization performance.

Table 2: **Impact of adaptive prompt weighting.**

| Base model | Promise variant | Base | New | H |
|---|---|---|---|---|
| MaPLe | w/o adaptive weighting | 82.93 | 75.87 | 79.24 |
| | w/ adaptive weighting | 83.57 | 77.26 | 80.29 |
| MMRL | w/o adaptive weighting | 85.97 | 77.53 | 81.53 |
| | w/ adaptive weighting | 86.14 | 78.84 | 82.33 |

**Benefit of Token-Specific Adaptive Learning Rate.** To isolate the effect of token-level adaptation, we compare three independent variants of our framework: (1) using a fixed learning rate for all tokens, (2) adopting a task-specific adaptive learning rate following MetaSGD (Li et al., 2017), and (3) our proposed token-specific learning rate module that assigns distinct learning rates to each prompt token. As shown in Table 3, the token-specific variant consistently yields better performance on both base and novel classes across MaPLe and MMRL.

**Table 4: Base-to-new generalization across 11 datasets.** Our method achieves the highest harmonic mean, demonstrating robust performance and superior novel class generalization compared to previous methods. Blue numbers indicate the best performance in each column, cyan highlights the second-best, and the values in parentheses represent the difference between our method and the best previous method in each column.

| ViT-B/16 | Base | Novel | H |
|---|---|---|---|
| CLIP (Radford et al., 2021) | 69.34 | 74.22 | 71.70 |
| CoCoOp (Zhou et al., 2022a) | 80.47 | 71.69 | 75.83 |
| PromptSRC (Khattak et al., 2023b) | 84.26 | 76.10 | 79.97 |
| UNIGRAM (Li et al., 2023) | 80.34 | 75.92 | 78.07 |
| MetaPrompt (Zhao et al., 2024) | 83.65 | 75.48 | 79.09 |
| CoPrompt (Roy & Etemad, 2024) | 84.00 | 77.23 | 80.48 |
| DePT (Zhang et al., 2024) | 85.19 | 76.17 | 80.43 |
| ProMetaR (Park et al., 2024) | 84.39 | 76.93 | 80.49 |
| HPT++ (Wang et al., 2024) | 84.13 | 77.99 | 80.95 |
| MMRL (Guo & Gu, 2025) | 85.68 | 77.16 | 81.20 |
| Ours | 86.14 (+0.46) | 78.84 (+0.85) | 82.33 (+1.13) |

**(a)** Average over 11 datasets.

| ViT-B/16 | Base | Novel | H |
|---|---|---|---|
| CLIP (Radford et al., 2021) | 72.43 | 68.14 | 70.22 |
| CoCoOp (Zhou et al., 2022a) | 75.98 | 70.43 | 73.10 |
| PromptSRC (Khattak et al., 2023b) | 77.60 | 70.73 | 74.01 |
| UNIGRAM (Li et al., 2023) | 76.60 | 70.69 | 73.53 |
| MetaPrompt (Zhao et al., 2024) | 77.52 | 70.83 | 74.02 |
| CoPrompt (Roy & Etemad, 2024) | 77.67 | 71.27 | 74.33 |
| DePT (Zhang et al., 2024) | 78.20 | 70.27 | 74.02 |
| ProMetaR (Park et al., 2024) | 77.76 | 70.75 | 74.09 |
| HPT++ (Wang et al., 2024) | 77.66 | 71.11 | 74.24 |
| MMRL (Guo & Gu, 2025) | 77.90 | 71.30 | 74.45 |
| Ours | 78.98 (+0.78) | 73.45 (+2.15) | 76.11 (+1.66) |

**(b)** ImageNet

| ViT-B/16 | Base | Novel | H |
|---|---|---|---|
| CLIP (Radford et al., 2021) | 96.84 | 94.00 | 95.40 |
| CoCoOp (Zhou et al., 2022a) | 97.96 | 93.81 | 95.84 |
| PromptSRC (Khattak et al., 2023b) | 98.10 | 94.03 | 96.02 |
| UNIGRAM (Li et al., 2023) | 98.07 | 95.11 | 96.57 |
| MetaPrompt (Zhao et al., 2024) | 98.13 | 94.58 | 96.32 |
| CoPrompt (Roy & Etemad, 2024) | 98.27 | 94.90 | 96.55 |
| DePT (Zhang et al., 2024) | 98.57 | 94.10 | 96.28 |
| ProMetaR (Park et al., 2024) | 98.11 | 94.29 | 96.16 |
| HPT++ (Wang et al., 2024) | 98.17 | 95.78 | 96.96 |
| MMRL (Guo & Gu, 2025) | 98.97 | 94.50 | 96.68 |
| Ours | 98.95 (-0.02) | 95.96 (+0.18) | 97.43 (+0.47) |

**(c)** Caltech101

| ViT-B/16 | Base | Novel | H |
|---|---|---|---|
| CLIP (Radford et al., 2021) | 91.17 | 97.26 | 94.12 |
| CoCoOp (Zhou et al., 2022a) | 95.20 | 97.69 | 96.43 |
| PromptSRC (Khattak et al., 2023b) | 95.33 | 97.30 | 96.30 |
| UNIGRAM (Li et al., 2023) | 94.94 | 97.94 | 96.42 |
| MetaPrompt (Zhao et al., 2024) | 95.53 | 97.00 | 96.26 |
| CoPrompt (Roy & Etemad, 2024) | 95.67 | 98.10 | 96.87 |
| DePT (Zhang et al., 2024) | 95.43 | 97.33 | 96.37 |
| ProMetaR (Park et al., 2024) | 95.57 | 97.57 | 96.49 |
| HPT++ (Wang et al., 2024) | 95.94 | 97.89 | 96.91 |
| MMRL (Guo & Gu, 2025) | 95.90 | 97.60 | 96.74 |
| Ours | 96.13 (+0.19) | 95.24 (-2.86) | 95.68 (-1.23) |

**(d)** OxfordPets

| ViT-B/16 | Base | Novel | H |
|---|---|---|---|
| CLIP (Radford et al., 2021) | 63.37 | 74.89 | 68.65 |
| CoCoOp (Zhou et al., 2022a) | 70.49 | 73.59 | 72.01 |
| PromptSRC (Khattak et al., 2023b) | 78.27 | 74.97 | 76.58 |
| UNIGRAM (Li et al., 2023) | 73.50 | 75.38 | 74.43 |
| MetaPrompt (Zhao et al., 2024) | 76.34 | 75.01 | 75.48 |
| CoPrompt (Roy & Etemad, 2024) | 76.97 | 74.40 | 75.66 |
| DePT (Zhang et al., 2024) | 80.80 | 75.00 | 77.79 |
| ProMetaR (Park et al., 2024) | 78.32 | 75.18 | 76.72 |
| HPT++ (Wang et al., 2024) | 76.99 | 74.24 | 75.59 |
| MMRL (Guo & Gu, 2025) | 81.30 | 75.07 | 78.06 |
| Ours | 83.15 (+1.85) | 79.94 (+4.56) | 81.51 (+3.45) |

**(e)** StanfordCars

| ViT-B/16 | Base | Novel | H |
|---|---|---|---|
| CLIP (Radford et al., 2021) | 72.08 | 77.80 | 74.83 |
| CoCoOp (Zhou et al., 2022a) | 94.87 | 71.75 | 81.71 |
| PromptSRC (Khattak et al., 2023b) | 98.07 | 76.50 | 85.95 |
| UNIGRAM (Li et al., 2023) | 95.20 | 76.21 | 84.65 |
| MetaPrompt (Zhao et al., 2024) | 97.66 | 74.49 | 84.52 |
| CoPrompt (Roy & Etemad, 2024) | 97.27 | 76.60 | 85.71 |
| DePT (Zhang et al., 2024) | 98.40 | 77.10 | 86.46 |
| ProMetaR (Park et al., 2024) | 98.13 | 77.66 | 86.70 |
| HPT++ (Wang et al., 2024) | 97.50 | 76.69 | 85.85 |
| MMRL (Guo & Gu, 2025) | 98.97 | 77.27 | 86.78 |
| Ours | 98.91 (-0.06) | 83.56 (+5.76) | 90.59 (+3.81) |

**(f)** Flowers102

| ViT-B/16 | Base | Novel | H |
|---|---|---|---|
| CLIP (Radford et al., 2021) | 90.10 | 91.22 | 90.66 |
| CoCoOp (Zhou et al., 2022a) | 90.70 | 91.29 | 90.99 |
| PromptSRC (Khattak et al., 2023b) | 90.67 | 91.53 | 91.10 |
| UNIGRAM (Li et al., 2023) | 90.84 | 92.12 | 91.48 |
| MetaPrompt (Zhao et al., 2024) | 90.74 | 91.85 | 91.29 |
| CoPrompt (Roy & Etemad, 2024) | 90.73 | 92.07 | 91.40 |
| DePT (Zhang et al., 2024) | 90.87 | 91.92 | 91.22 |
| ProMetaR (Park et al., 2024) | 90.80 | 91.89 | 91.34 |
| HPT++ (Wang et al., 2024) | 90.56 | 91.62 | 91.09 |
| MMRL (Guo & Gu, 2025) | 90.57 | 91.50 | 91.03 |
| Ours | 91.21 (+0.34) | 93.16 (+1.04) | 92.17 (+0.69) |

**(g)** Food101

| ViT-B/16 | Base | Novel | H |
|---|---|---|---|
| CLIP (Radford et al., 2021) | 27.19 | 36.29 | 31.09 |
| CoCoOp (Zhou et al., 2022a) | 33.41 | 23.71 | 27.74 |
| PromptSRC (Khattak et al., 2023b) | 42.73 | 37.87 | 40.15 |
| UNIGRAM (Li et al., 2023) | 32.25 | 38.00 | 34.89 |
| MetaPrompt (Zhao et al., 2024) | 40.14 | 36.51 | 38.24 |
| CoPrompt (Roy & Etemad, 2024) | 40.20 | 39.33 | 39.76 |
| DePT (Zhang et al., 2024) | 45.70 | 36.73 | 40.73 |
| ProMetaR (Park et al., 2024) | 42.02 | 38.63 | 40.25 |
| HPT++ (Wang et al., 2024) | 40.50 | 42.19 | 41.33 |
| MMRL (Guo & Gu, 2025) | 46.30 | 37.03 | 41.15 |
| Ours | 46.77 (+0.47) | 41.75 (-0.44) | 44.12 (+2.79) |

**(h)** FGVCAircraft

| ViT-B/16 | Base | Novel | H |
|---|---|---|---|
| CLIP (Radford et al., 2021) | 69.36 | 75.35 | 72.23 |
| CoCoOp (Zhou et al., 2022a) | 79.74 | 76.86 | 78.27 |
| PromptSRC (Khattak et al., 2023b) | 82.67 | 78.47 | 80.52 |
| UNIGRAM (Li et al., 2023) | 80.43 | 77.91 | 79.15 |
| MetaPrompt (Zhao et al., 2024) | 82.26 | 79.04 | 80.62 |
| CoPrompt (Roy & Etemad, 2024) | 82.63 | 80.03 | 81.31 |
| DePT (Zhang et al., 2024) | 83.27 | 78.97 | 81.06 |
| ProMetaR (Park et al., 2024) | 82.70 | 79.02 | 80.82 |
| HPT++ (Wang et al., 2024) | 82.40 | 79.86 | 81.11 |
| MMRL (Guo & Gu, 2025) | 83.20 | 79.30 | 81.20 |
| Ours | 83.23 (-0.04) | 79.58 (-0.45) | 81.36 (+0.05) |

**(i)** SUN397

| ViT-B/16 | Base | Novel | H |
|---|---|---|---|
| CLIP (Radford et al., 2021) | 53.24 | 59.90 | 56.37 |
| CoCoOp (Zhou et al., 2022a) | 77.01 | 56.00 | 64.85 |
| PromptSRC (Khattak et al., 2023b) | 83.37 | 62.97 | 71.75 |
| UNIGRAM (Li et al., 2023) | 73.62 | 62.38 | 67.56 |
| MetaPrompt (Zhao et al., 2024) | 83.10 | 58.05 | 68.35 |
| CoPrompt (Roy & Etemad, 2024) | 83.13 | 64.73 | 72.79 |
| DePT (Zhang et al., 2024) | 84.80 | 61.20 | 71.09 |
| ProMetaR (Park et al., 2024) | 83.02 | 64.05 | 72.31 |
| HPT++ (Wang et al., 2024) | 84.18 | 66.39 | 74.23 |
| MMRL (Guo & Gu, 2025) | 85.67 | 65.00 | 73.82 |
| Ours | 86.25 (+0.58) | 67.97 (+1.58) | 76.03 (+1.80) |

**(j)** DTD

| ViT-B/16 | Base | Novel | H |
|---|---|---|---|
| CLIP (Radford et al., 2021) | 56.48 | 64.05 | 60.03 |
| CoCoOp (Zhou et al., 2022a) | 87.49 | 60.04 | 71.21 |
| PromptSRC (Khattak et al., 2023b) | 92.90 | 73.90 | 82.32 |
| UNIGRAM (Li et al., 2023) | 86.26 | 71.38 | 78.12 |
| MetaPrompt (Zhao et al., 2024) | 93.53 | 75.21 | 83.38 |
| CoPrompt (Roy & Etemad, 2024) | 94.60 | 78.57 | 85.84 |
| DePT (Zhang et al., 2024) | 93.23 | 77.90 | 84.88 |
| ProMetaR (Park et al., 2024) | 94.94 | 77.44 | 85.30 |
| HPT++ (Wang et al., 2024) | 95.31 | 80.64 | 87.36 |
| MMRL (Guo & Gu, 2025) | 95.60 | 80.17 | 87.21 |
| Ours | 95.41 (-0.19) | 77.64 (-3.00) | 85.61 (-1.75) |

**(k)** EuroSAT

| ViT-B/16 | Base | Novel | H |
|---|---|---|---|
| CLIP (Radford et al., 2021) | 70.53 | 77.50 | 73.85 |
| CoCoOp (Zhou et al., 2022a) | 82.33 | 73.45 | 77.64 |
| PromptSRC (Khattak et al., 2023b) | 87.10 | 78.80 | 82.74 |
| UNIGRAM (Li et al., 2023) | 82.00 | 78.06 | 79.98 |
| MetaPrompt (Zhao et al., 2024) | 85.33 | 77.72 | 81.35 |
| CoPrompt (Roy & Etemad, 2024) | 86.90 | 79.57 | 83.07 |
| DePT (Zhang et al., 2024) | 87.73 | 77.90 | 82.46 |
| ProMetaR (Park et al., 2024) | 86.97 | 79.84 | 83.25 |
| HPT++ (Wang et al., 2024) | 86.26 | 81.50 | 83.81 |
| MMRL (Guo & Gu, 2025) | 88.10 | 80.07 | 83.89 |
| Ours | 88.59 (+0.49) | 78.99 (-2.51) | 83.52 (-0.37) |

**(l)** UCF101

These results confirm the benefit of learning fine-grained, data-driven adaptation rates within the inner loop, providing more precise control over token optimization than task-level adaptation alone.

**Table 3: Effect of token-specific adaptive learning rate.**

| Base model | Promise variant | Base | New | H |
|---|---|---|---|---|
| MaPLe | w/o token-specific adaptive learning rate | 82.45 | 76.01 | 79.08 |
| | w/ task-specific adaptive learning rate | 82.93 | 76.29 | 79.25 |
| | w/ token-specific adaptive learning rate | 83.57 | 77.26 | 80.29 |
| MMRL | w/o token-specific adaptive learning rate | 85.79 | 77.42 | 81.39 |
| | w/ task-specific adaptive learning rate | 85.99 | 78.17 | 81.48 |
| | w/ token-specific adaptive learning rate | 86.14 | 78.84 | 82.33 |

**Robustness–cost trade-off.** Promise is most useful in scenarios where robustness to prompt variation matters, not just raw accuracy. In many VLM applications, a model that behaves stably under different user phrasings can be more valuable than one that gains a small amount of accuracy but remains highly sensitive to wording. In terms of efficiency, Promise only updates the soft prompt parameters and the small weighting / preconditioning modules; the vision and text encoders remain frozen. At inference time, the computation graph is identical to the underlying prompt learner, so the runtime and memory cost at deployment are unchanged. The additional cost appears only during training. In our implementation, meta-finetuning increases wall-clock training time by about 18–22% compared with standard prompt tuning for the same number of epochs, mainly due to the extra outer-loop gradient and token-wise preconditioning. For example, on ImageNet with MaPLe on a single A6000, vanilla training takes about 3.4 hours, while Promise takes about 4.0 hours; on smaller datasets, the absolute overhead is correspondingly smaller. We also experimented with a lighter variant that removes the token-wise learning-rate module and keeps only adaptive weighting; this reduces the overhead by roughly 8–10%, but also weakens the robustness gains. Since meta-finetuning is a one-time offline stage and inference cost is unchanged, we consider this training overhead acceptable in settings where prompts are reused extensively and stable behavior under prompt shifts is important.

**Base-to-New Generalization.** To evaluate the generalization performance of our method on novel classes, we conduct base-to-new experiments across 11 diverse datasets. As shown in Table 4,

**Table 5: Cross-dataset generalization.** Our method consistently improves cross-dataset generalization. Blue numbers indicate the best performance in each column, cyan highlights the second-best, and the values in parentheses represent the difference between our method and the best previous method in each column.

| | Source | Target | | | | | | | | | | |
|---|---|---|---|---|---|---|---|---|---|---|---|---|
| | ImageNet | Caltech101 | OxfordPets | StanfordCars | Flowers102 | Food101 | Aircraft | SUN397 | DTD | EuroSAT | UCF101 | Average |
| CLIP (Radford et al., 2021) | 71.51 | 93.70 | 89.14 | 64.51 | 68.71 | 85.30 | 18.47 | 64.15 | 41.92 | 46.39 | 66.55 | 63.88 |
| CoOp (Zhou et al., 2022b) | 71.51 | 93.70 | 89.14 | 64.51 | 68.71 | 85.30 | 18.47 | 64.15 | 41.92 | 46.39 | 66.55 | 63.88 |
| CoCoOp (Zhou et al., 2022a) | 71.02 | 94.43 | 90.14 | 65.32 | 71.88 | 86.06 | 22.94 | 67.36 | 45.73 | 45.37 | 68.21 | 65.74 |
| PromptSRC (Khattak et al., 2023b) | 71.27 | 93.60 | 90.25 | 65.70 | 70.25 | 86.15 | 23.90 | 67.10 | 46.87 | 45.50 | 68.75 | 65.81 |
| MetaPrompt (Zhao et al., 2024) | 71.27 | 93.60 | 90.25 | 65.70 | 70.25 | 86.15 | 23.90 | 67.10 | 46.87 | 45.50 | 68.75 | 65.81 |
| DePT (Zhang et al., 2024) | 71.60 | 93.80 | 90.13 | 66.00 | 70.93 | 86.27 | 24.30 | 67.23 | 46.60 | 45.83 | 69.10 | 66.02 |
| ProMetaR (Park et al., 2024) | 71.29 | 93.74 | 90.59 | 65.83 | 71.13 | 86.39 | 24.78 | 67.41 | 47.08 | 45.02 | 69.50 | 66.15 |
| MaPLe (Khattak et al., 2023a) | 70.72 | 93.53 | 90.49 | 65.57 | 72.23 | 86.20 | 24.74 | 67.01 | 46.49 | 48.06 | 68.69 | 66.30 |
| ATPrompt Li et al. (2025) | 70.69 | 94.04 | 91.03 | 66.06 | 71.99 | 86.33 | 24.42 | 67.05 | 45.21 | 48.63 | 69.15 | 66.75 |
| CoPrompt (Roy & Etemad, 2024) | 70.80 | 94.50 | 90.73 | 65.67 | 72.30 | 86.43 | 24.00 | 67.57 | 47.07 | 51.90 | 69.73 | 67.00 |
| FedMVP (Singha et al., 2025) | 70.87 | 95.37 | 89.27 | 65.83 | 72.80 | 87.06 | 25.94 | 68.19 | 49.78 | 50.84 | 70.58 | 67.57 |
| MMRL (Guo & Gu, 2025) | 72.03 | 94.67 | 91.43 | 66.10 | 72.77 | 86.40 | 26.30 | 67.57 | 45.90 | 53.10 | 68.27 | 67.25 |
| HiCroPL (Zheng et al., 2025) | 70.84 | 94.48 | 90.13 | 65.68 | 72.03 | 86.46 | 26.58 | 68.78 | 53.19 | 49.19 | 70.31 | 67.68 |
| HPT++ (Wang et al., 2024) | 71.81 | 94.02 | 92.16 | 65.55 | 72.43 | 86.34 | 28.60 | 68.78 | 51.02 | 50.76 | 70.53 | 68.02 |
| **Ours** | **72.77** | **95.72** | **92.73** | **67.09** | **73.56** | **86.95** | 28.15 | **68.94** | 47.92 | **54.21** | **70.62** | **68.68** |
| | (+0.74) | (+0.35) | (+0.57) | (+0.99) | (+0.76) | (-0.11) | (-0.45) | (+0.16) | (-5.27) | (+1.11) | (+0.04) | (+0.66) |

our method is directly compared with the strongest existing prompt learning approaches. Across all datasets, our method consistently achieves the highest harmonic mean, indicating balanced performance on both base and novel classes. On average, it surpasses the best-performing baseline (MMRL) by 1.13 points in H. In challenging fine-grained benchmarks such as FGVCAircraft and Flowers102, our method shows clear advantages—improving H by 2.79 and 3.81 points respectively compared to the strongest alternative. Overall, these results indicate that our prompt-robust meta-finetuning framework offers modest but consistent gains in accuracy, and, together with the reduced sensitivity to prompt phrasing observed in our robustness analysis, leads to more stable base-to-new performance in both coarse- and fine-grained domains.

**Cross-dataset Generalization.** To evaluate the robustness of our method under domain shift, we conduct cross-dataset generalization experiments following the standard protocol. As shown in Table 5, our method outperforms all competing approaches across different datasets. These results highlight the ability of our prompt-robust meta-learning framework to generalize effectively across unseen distributions and confirm its superiority in cross-domain prompt transfer.

**Domain Generalization.** We evaluate the domain generalization ability of our method using the standard ImageNet robustness benchmark, where models trained on ImageNet are tested on four shifted domains: ImageNet-V2, -S, -A, and -R. As shown in Table 6, our method outperforms all competing prompt learning methods, achieving the highest average accuracy. Ours yields strong improvements across all domain variants, notably outperforming HPT++ on the more challenging shifts such as ImageNet-A and -R. These results confirm that our prompt-robust design not only improves in-domain generalization but also significantly enhances resilience to distributional shifts.

**Table 6: Domain generalization.**

| | Source | Target | | | | |
|---|---|---|---|---|---|---|
| | ImageNet | -V2 | -S | -A | -R | Avg. |
| CLIP (Radford et al., 2021) | 66.73 | 60.83 | 46.15 | 47.77 | 73.96 | 57.17 |
| CoOp (Zhou et al., 2022b) | 71.51 | 64.20 | 47.99 | 49.71 | 75.21 | 59.28 |
| CoCoOp (Zhou et al., 2022a) | 71.02 | 64.07 | 48.75 | 50.63 | 76.18 | 59.90 |
| MaPLe (Khattak et al., 2023a) | 70.72 | 64.07 | 49.15 | 50.90 | 76.98 | 60.27 |
| ATPrompt (Li et al., 2025) | 70.69 | 64.40 | 49.10 | 51.77 | 77.11 | 60.60 |
| CoPrompt (Roy & Etemad, 2024) | 70.80 | 64.25 | 49.43 | 50.50 | 77.51 | 60.42 |
| HiCroPL (Zheng et al., 2025) | 71.22 | 64.33 | 49.47 | 50.79 | 77.15 | 60.44 |
| MMRL (Guo & Gu, 2025) | 72.03 | 64.47 | 49.17 | 51.20 | 77.53 | 60.59 |
| PromptSRC (Khattak et al., 2023b) | 71.27 | 64.35 | 49.55 | 50.90 | 77.80 | 60.65 |
| ProMetaR (Park et al., 2024) | 71.29 | 64.39 | 49.55 | 51.25 | 77.89 | 60.77 |
| FedMVP (Singha et al., 2025) | 70.87 | 63.72 | 50.93 | 51.76 | 77.23 | 60.91 |
| HPT++ (Wang et al., 2024) | 71.81 | 65.31 | 49.28 | 51.18 | 77.52 | 60.82 |
| **Ours** | **72.77** | **66.92** | 50.28 | **53.21** | **78.52** | **62.23** |
| | (+0.74) | (+1.61) | (-0.65) | (+1.44) | (+0.63) | (+1.32) |

## 6 CONCLUSION

We presented **Promise**, a meta-learning framework for prompt-robust vision–language modeling that learns to generalize across diverse prompt formulations. **Promise** combines a dual-loop adaptation scheme with adaptive prompt weighting and token-specific learning rates, enabling fine-grained, context-aware prompt optimization. Beyond empirical gains, our analysis shows that the weighted–preconditioned inner update induces a single-step decrease of the outer empirical risk while contracting across-prompt sensitivity and tightens a data-dependent generalization bound at the post-inner initialization. Experiments on base-to-new generalization, cross-dataset transfer, and domain-shift benchmarks corroborate these guarantees, yielding consistent improvements over state-of-the-art prompt tuning methods.

**Ethics Statement** This work introduces **Promise**, a meta-finetuning framework to improve prompt robustness of vision–language models. We do not collect new human data or annotate personal information; all experiments use *public* datasets under their original licenses (e.g., ImageNet variants and standard cross-dataset/domain-shift benchmarks). Our method does not attempt to infer demographics, identities, or other sensitive attributes. Nevertheless, adapted models may inherit societal biases present in pretraining/evaluation data and could be misused for privacy-invasive applications (e.g., surveillance on personal images) or for processing proprietary content without authorization. We discourage such uses and recommend adherence to data-governance policies, license terms, and applicable laws. We include dataset/usage licenses, prompt lists, and evaluation details to support responsible replication; we also report compute profiles to increase transparency about environmental impact.

**Reproducibility Statement** All models, datasets, and baselines used in this paper are publicly accessible. We specify training/validation splits, optimization settings, and evaluation protocols in the main text, with additional implementation details in the appendix. To facilitate replication: (i) we provide the complete prompt templates and structured-prompt variants; (ii) we include an algorithmic description with pseudocode and complexity notes; (iii) we report hardware, wall-clock training time, and memory usage; and (iv) we add qualitative visualizations to verify robustness behaviors. We release configuration files, random seeds, and checkpoints together with evaluation scripts to exactly reproduce the reported numbers.

**Acknowledgments** Our work was supported by the Research Start-up fund from the Department of Statistics and Data Science, Beijing Normal-Hong Kong Baptist University; the Guangdong Provincial Key Laboratory of IRADS (2022B1212010006); the Guangdong Higher Education Upgrading Plan (2021–2025); and the Guangdong and Hong Kong Universities "1+1+1" Joint Research Collaboration Scheme (2025A050500004).

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

## A  LLM Usage Statement

We used a large language model (ChatGPT) solely for grammar checking and language polishing of the manuscript text. It did not contribute to research ideation, method design, experiments, data analysis, or result generation; all technical content was authored and verified by the authors.

## B  Proofs and Auxiliary Results

**Notation.**  Write the outer empirical risk as

$$R(\theta) := \widehat{R}_{P_{\text{out}}}(\theta) = \frac{1}{|P_{\text{out}}|} \sum_{T \in P_{\text{out}}} L(\theta; T),$$

and the (weighted) inner aggregated gradient as

$$G(\theta) := \sum_{T \in P_{\text{in}}} w_T \, \nabla_\theta L(\theta; T), \qquad \mu(\theta) := \mathbb{E}[G(\theta)], \qquad \Sigma_w := \text{Var}[G(\theta)].$$

The post-inner update is $\hat{\theta} = \theta - P \, G(\theta)$ with a diagonal $P = \text{diag}(\alpha_1, \ldots, \alpha_d) \succeq 0$.

### B.1  Assumptions

**A0: Disjoint and independent draws.** $P_{\text{in}}$ and $P_{\text{out}}$ are drawn independently from $\mathcal{P}$ and $P_{\text{in}} \cap P_{\text{out}} = \emptyset$.

**A1:  Smoothness.**  For every prompt $T$, $L(\cdot; T)$ is $L$-smooth; hence $R(\cdot)$ is also $L$-smooth: $\|\nabla L(\theta; T) - \nabla L(\theta'; T)\| \leq L \|\theta - \theta'\|$, $\|\nabla R(\theta) - \nabla R(\theta')\| \leq L \|\theta - \theta'\|$.

**A2; Unbiased inner gradient with bounded second moment.** $\mathbb{E}[G(\theta)] = \mu(\theta)$ and $\mathbb{E}\|G(\theta) - \mu(\theta)\|^2 = \text{Tr}(\Sigma_w) \leq \sigma^2$.

**A3: Preconditioner.** $P = P^\top \succeq 0$ is diagonal, with spectral bounds $0 \leq \lambda_{\min}(P) \leq \|P\|_2$.

**A4: Alignment and bounded amplification.** There exist $\gamma \in (0, 1]$ and $\kappa \geq 1$ such that

$$\langle \nabla R(\theta), \, P \, \mu(\theta) \rangle \, \geq \, \gamma \, \lambda_{\min}(P) \, \|\nabla R(\theta)\|^2, \qquad \|\mu(\theta)\| \, \leq \, \kappa \, \|\nabla R(\theta)\|. \tag{5}$$

**A5: Lipschitz properties.**  (i) (*Model Lipschitz for sensitivity*) There exists $\Gamma_f > 0$ such that $|f_\theta(x; T) - f_{\theta'}(x; T)| \leq \Gamma_f \|\theta - \theta'\|$ for all $(x, T)$. (ii) (*Loss Lipschitz in parameters*) There exists $\Gamma_L > 0$ such that $|L(\theta; T) - L(\theta'; T)| \leq \Gamma_L \|\theta - \theta'\|$ for all $T$.

**A6: Prompt-coherence for sensitivity.** Let $S(\theta) := \text{Var}_{T \sim \mathcal{P}}[f_\theta(x; T)]$ (or its Jacobian surrogate). There exist $\mu_{\text{eff}} \in (0, 1]$ and $C_0 \geq 0$ such that

$$S(\theta - P \, \mu(\theta)) \, \leq \, (1 - \mu_{\text{eff}}) \, S(\theta) \, + \, C_0 \, \|P \, \mu(\theta)\|^2. \tag{6}$$

### B.2  Auxiliary Lemmas

**Lemma B.1** (Smoothness descent). *For any $v \in \mathbb{R}^d$, $R(\theta - Pv) \leq R(\theta) - \langle \nabla R(\theta), Pv \rangle + \frac{L}{2} \|Pv\|^2$.*

*Proof.* By $L$-smoothness, $R(y) \leq R(x) + \langle \nabla R(x), y - x \rangle + \frac{L}{2} \|y - x\|^2$. Set $x = \theta$ and $y = \theta - Pv$. $\square$

**Lemma B.2** (Second moment under preconditioning). $\mathbb{E}\|P \, G(\theta)\|^2 \, \leq \, \|P\|_2^2 (\|\mu(\theta)\|^2 + \text{Tr}(\Sigma_w))$.

*Proof.* Write $G = \mu + \xi$ with $\mathbb{E}[\xi] = 0$, $\text{Cov}(\xi) = \Sigma_w$. Then $\mathbb{E}\|PG\|^2 = \|P\mu\|^2 + \mathbb{E}\|P\xi\|^2 \leq \|P\|_2^2 \|\mu\|^2 + \|P\|_2^2 \text{Tr}(\Sigma_w)$. $\square$

## B.3 PROOF OF THEOREM 4.1

Recall $\hat{\theta} = \theta - P\,G(\theta)$. Applying Lemma B.1 with $v = G(\theta)$ and taking expectation over $P_{\text{in}}$,

$$
\begin{aligned}
\mathbb{E}\,R(\hat{\theta}) &\leq R(\theta) \;-\; \big\langle \nabla R(\theta),\, P\,\mu(\theta) \big\rangle \;+\; \tfrac{L}{2}\,\mathbb{E}\|P\,G(\theta)\|^2 \\
&\leq R(\theta) \;-\; \gamma\,\lambda_{\min}(P)\,\|\nabla R(\theta)\|^2 \;+\; \tfrac{L}{2}\|P\|_2^2\Big(\|\mu(\theta)\|^2 + \text{Tr}(\Sigma_w)\Big) \\
&\leq R(\theta) \;-\; \Big(\gamma\,\lambda_{\min}(P) - \tfrac{L}{2}\kappa^2\|P\|_2^2\Big)\|\nabla R(\theta)\|^2 \;+\; \tfrac{L}{2}\|P\|_2^2\,\text{Tr}(\Sigma_w),
\end{aligned}
$$

where the second line uses Lemma B.2 and the alignment bound in equation 5, and the third line uses $\|\mu(\theta)\| \leq \kappa\|\nabla R(\theta)\|$. If $\gamma\,\lambda_{\min}(P) > \tfrac{L}{2}\kappa^2\|P\|_2^2$, the expectation strictly decreases.

*Sensitivity contraction.* Let $\bar{\theta} = \theta - P\,\mu(\theta)$ be the noise-free step. By equation 6, $S(\bar{\theta}) \leq (1 - \mu_{\text{eff}})S(\theta) + C_0\|P\,\mu(\theta)\|^2$. Note that $\hat{\theta} - \bar{\theta} = -P\,(G - \mu)$. By the Lipschitz property of $f_\theta$ in A5(i) and the definition of variance,

$$
\mathbb{E}\,S(\hat{\theta}) \;\leq\; S(\bar{\theta}) \;+\; \Gamma_f^2\,\mathbb{E}\|P(G - \mu)\|^2 \;\leq\; (1 - \mu_{\text{eff}})S(\theta) + C_0\|P\mu\|^2 + \Gamma_f^2\|P\|_2^2\,\text{Tr}(\Sigma_w).
$$

Using $\|P\mu\|^2 \leq \|P\|_2^2\,\kappa^2\|\nabla R(\theta)\|^2$ and absorbing constants into $C$ yields $\mathbb{E}S(\hat{\theta}) \leq (1 - \mu_{\text{eff}})S(\theta) + C\,\|P\|_2^2\,\text{Tr}(\Sigma_w)$. $\qquad\square$

**Remark.** If one subsequently takes an outer gradient step $\theta^+ = \hat{\theta} - \eta\nabla R(\hat{\theta})$ with $\eta \in \big(0,\, 1/(L\|P\|_2)\big)$, then by $L$-smoothness, $R(\theta^+) \leq R(\hat{\theta}) - \eta\|\nabla R(\hat{\theta})\|^2 + \tfrac{L}{2}\eta^2\|\nabla R(\hat{\theta})\|^2$, which further decreases the risk for suitably small $\eta$.

## B.4 PROOF OF THEOREM 4.2

Decompose the error into (i) *initialization noise* from the inner step, and (ii) *sampling error* from $P_{\text{out}}$.

**Step 1: Initialization noise bound.** Write $\hat{\theta} = \theta - P(\mu + \xi)$ with $\mathbb{E}[\xi] = 0$ and $\text{Cov}(\xi) = \Sigma_w$. By A5(ii) (loss Lipschitz in $\theta$) and the triangle inequality,

$$
\big|R_{\text{pop}}(\hat{\theta}) - R_{\text{pop}}(\theta - P\mu)\big| \;\leq\; \Gamma_L\,\|\hat{\theta} - (\theta - P\mu)\| \;=\; \Gamma_L\,\|P\xi\|.
$$

Taking expectation and using Jensen, $\mathbb{E}\big[R_{\text{pop}}(\hat{\theta})\big] \leq R_{\text{pop}}(\theta - P\mu) + \Gamma_L\,\sqrt{\mathbb{E}\|P\xi\|^2} \leq R_{\text{pop}}(\theta - P\mu) + \Gamma_L\,\sqrt{\text{Tr}(P\Sigma_w P)}$.

**Step 2: Concentration on $P_{\text{out}}$.** Conditioned on $\hat{\theta}$, by A0 the samples in $P_{\text{out}}$ are i.i.d. If $L \in [0, 1]$ (or is sub-Gaussian after scaling), Hoeffding's inequality gives, for any $\delta \in (0, 1)$, with probability at least $1 - \delta$,

$$
R_{\text{pop}}(\hat{\theta}) \;\leq\; R(\hat{\theta}) \;+\; c_2\sqrt{\frac{\ln(1/\delta)}{n_{\text{out}}}},
$$

where $n_{\text{out}}$ is the number of samples in $\widehat{R}_{P_{\text{out}}}$ and $c_2 > 0$ is an absolute constant. Combining Step 1 and the concentration of $R(\hat{\theta})$ around $R_{\text{pop}}(\hat{\theta})$, we obtain

$$
R_{\text{pop}}(\hat{\theta}) \;\leq\; R(\hat{\theta}) \;+\; c_2\sqrt{\frac{\ln(1/\delta)}{n_{\text{out}}}} \;+\; c_1\,\Gamma_L\,\sqrt{\text{Tr}(P\Sigma_w P)},
$$

after absorbing universal constants into $c_1 > 0$. This yields the stated bound. $\qquad\square$

## B.5 COROLLARY AND PRACTICAL DIAGNOSTICS

**Corollary B.3** (Improvement over uniform inner update). *Let $\hat{\theta}_{\text{uni}} = \theta - \alpha\frac{1}{|P_{\text{in}}|}\sum_{T \in P_{\text{in}}} \nabla_\theta L(\theta; T)$. If there exist $(w_T, P)$ such that the alignment and amplification in equation 5 hold with $\gamma\,\lambda_{\min}(P) > \frac{L}{2}\kappa^2\|P\|_2^2$ and $\text{Tr}(P\Sigma_w P) \leq \text{Tr}(P\Sigma_{\text{uni}} P)$, then $\mathbb{E}\,R(\hat{\theta}) \leq \mathbb{E}\,R(\hat{\theta}_{\text{uni}})$ and the variance term in Theorem 4.2 is smaller for $(w_T, P)$, resulting in a strictly tighter bound.*

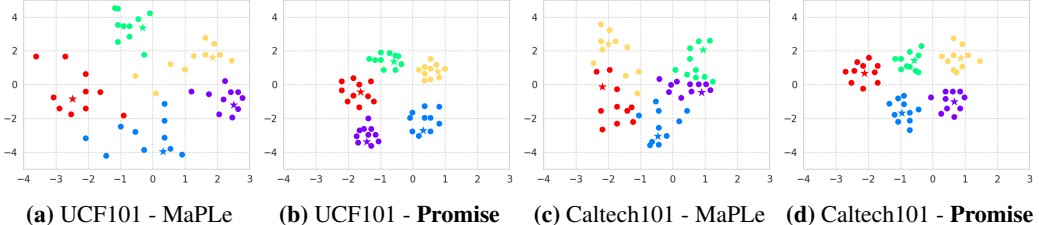

**(a)** UCF101 - MaPLe    **(b)** UCF101 - **Promise**    **(c)** Caltech101 - MaPLe    **(d)** Caltech101 - **Promise**

**Figure 4: Visualization of prompt sensitivity and robustness with unseen prompts.** Each color denotes a class. Stars are image embeddings; circles are text embeddings from ten unseen but semantically equivalent prompts generated by GPT-4o. MaPLe shows high prompt sensitivity (large star–circle gaps), while our method (**Promise**) produces more consistent, tightly clustered embeddings.

**Table 7: Effect of structures prompt on Promise.**

|  | Base | New | H |
|---|---|---|---|
| MaPLe (Khattak et al., 2023a) | 82.28 | 75.14 | 78.55 |
| + **Promise** (CLIP prompts) | 83.57 | 77.26 | 80.29 |
| + **Promise** (LLM prompts) | 84.21 | 79.42 | 81.74 |

**Diagnostics.** (i) *Alignment.* Track $\cos \angle\big(\nabla R(\theta),\, P\,G(\theta)\big)$ as an empirical lower bound for $\gamma$; (ii) *Variance term.* Report $\mathrm{Tr}(P\widehat{\Sigma}_w P)$ with batch-level estimates $\widehat{\Sigma}_w$ across training; (iii) *Sensitivity.* Monitor $S(\theta)$ (APV or Jacobian surrogate) and its contraction to validate equation 6.

## C   VISUALIZATION OF PROMPT ROBUSTNESS

To further examine the robustness of our method to prompt variations, we visualize the feature distributions obtained from different prompt templates using t-SNE (Van der Maaten & Hinton, 2008), as shown in Figure 4. Each class is represented by a unique color; stars denote the image embeddings, and circles correspond to text embeddings generated from ten semantically equivalent but unseen prompts using GPT-4o.

Compared to MaPLe, which exhibits high sensitivity to prompt variations—evidenced by the wide spread of circles around each star—our method, **Promise**, produces tightly clustered embeddings. This indicates that our method yields more stable and consistent text representations across diverse prompt formulations. Such consistency demonstrates **Promise**'s effectiveness in mitigating prompt sensitivity and achieving prompt-agnostic behavior.

## D   EFFECT OF STRUCTURED PROMPTS.

While our main experiments employ handcrafted CLIP-style prompt templates to simulate realistic prompt variation, recent advances suggest that prompts generated by LLMs can provide greater semantic diversity and syntactic richness. To evaluate whether our proposal can benefit from such structured prompt generation, we adopt the IPO framework (Du et al., 2024), which uses LLMs to automatically generate 80 diverse and interpretable prompt templates. We retrain **Promise** using the IPO-generated prompts, keeping all other experimental settings unchanged. As shown in Table 7, incorporating LLM-generated prompts leads to consistent performance improvements across both base and novel classes. These results confirm that **Promise** not only maintains compatibility with structured prompts but also benefits from their expanded distributional coverage.

## E   DETAILED PROMPTS

To implement the textual diversity technique, we randomly select 60 prompt templates from the full list of templates provided in Radford et al. (2021). Specifically, the following prompt templates are utilized in our textual diversity component.

```
"a photo of a {CLASS}."
"a bad photo of a {CLASS}."
"a photo of many {CLASS}."
"a sculpture of a {CLASS}."
"a photo of the hard to see {CLASS}."
"a low resolution photo of the {CLASS}."
"a rendering of a {CLASS}."
"graffiti of a {CLASS}."
"a bad photo of the {CLASS}."
"a cropped photo of the {CLASS}."
"a tattoo of a {CLASS}."
"the embroidered {CLASS}."
"a photo of a hard to see {CLASS}."
"a bright photo of a {CLASS}."
"a photo of a clean {CLASS}."
"a photo of a dirty {CLASS}."
"a dark photo of the {CLASS}."
"a drawing of a {CLASS}."
"a photo of my {CLASS}."
"the plastic {CLASS}."
"a photo of the cool {CLASS}."
"a close-up photo of a {CLASS}."
"a black and white photo of the {CLASS}."
"a painting of the {CLASS}."
"a painting of a {CLASS}."
"a pixelated photo of the {CLASS}."
"a sculpture of the {CLASS}."
"a bright photo of the {CLASS}."
"a cropped photo of a {CLASS}."
"a plastic {CLASS}."
"a photo of the dirty {CLASS}."
"a jpeg corrupted photo of a {CLASS}."
"a blurry photo of the {CLASS}."
"a photo of the {CLASS}."
"a good photo of the {CLASS}."
"a rendering of the {CLASS}."
"a {CLASS} in a video game."
"a photo of one {CLASS}."
"a doodle of a {CLASS}."
"a close-up photo of the {CLASS}."
"the origami {CLASS}."
"the {CLASS} in a video game."
"a sketch of a {CLASS}."
"a doodle of the {CLASS}."
"an origami {CLASS}."
"a low resolution photo of a {CLASS}."
"the toy {CLASS}."
"a rendition of the {CLASS}."
"a photo of the clean {CLASS}."
"a photo of a large {CLASS}."
"a rendition of a {CLASS}."
"a photo of a nice {CLASS}."
"a photo of a weird {CLASS}."
"a blurry photo of a {CLASS}."
"a cartoon {CLASS}."
"art of a {CLASS}."
"a sketch of the {CLASS}."
"a embroidered {CLASS}."
```

```
"a pixelated photo of a {CLASS}."
"itap of the {CLASS}."
```

## F  ALGORITHM DESCRIPTION

The **Promise** algorithm, outlined in Algorithm 1, is designed to improve prompt learning for zero-shot generalization. It employs a meta-learning approach with two loops: the inner loop for task-specific adaptation and the outer loop for optimizing meta-parameters for generalization across diverse prompts. Key components of the algorithm include: (1) **Inner Loop**: The inner loop adapts the parameters of the CLIP model $\theta$ using a sub-set of prompts $\mathcal{P}_{\text{in}}$ and data $\mathcal{D}_{\text{in}}$. A hypernetwork $H$ dynamically generates learning rates for prompt-specific updates, enabling efficient optimization. The adapted parameters $\hat{\theta}$ are obtained after several gradient steps. (2) **Outer Loop**: The outer loop aims to improve the generalization capability of the meta-parameters $\theta$ using a distinct subset of prompts $\mathcal{P}_{\text{out}}$ and data $\mathcal{D}_{\text{out}}$. The meta-parameters $\theta$, hypernetwork $H$, and prompt weights $\{w_i\}$ are updated by minimizing the loss over $\mathcal{P}_{\text{out}}$, ensuring robustness across varied prompts. (3) **Prompt Weight Adjustment**: To balance the contribution of different prompts, the weights $\{w_i\}$ are updated using an exponential normalization mechanism based on the gradient of the outer-loop loss. This iterative process ensures that **Promise** learns to generalize across diverse prompt templates, enhancing the zero-shot capabilities of the underlying vision-language model. By leveraging first-order approximations in the outer loop, inspired by Reptile, the algorithm avoids computational overhead from second-order gradients, making it more time-efficient.

---

**Algorithm 1 Promise**

**Input:** Dataset $\mathcal{D}$, CLIP parameters $\theta = \{\theta_t, \theta_v\}$, Prompt set $\mathcal{P} = \{\mathbf{T}_1, \mathbf{T}_2, \ldots, \mathbf{T}_M\}$, learning rates $\alpha, \beta$.
**Require:** Inner-loop data $\mathcal{D}_{\text{in}}, \mathcal{P}_{\text{in}}$, Outer-loop data $\mathcal{D}_{\text{out}}, \mathcal{P}_{\text{out}}$, Hypernetwork $H(\cdot)$, weights $\{w_i\}$.
**Ensure** $\mathcal{P}_{\text{in}} \cap \mathcal{P}_{\text{out}} = \emptyset, \mathcal{D}_{\text{in}} \cap \mathcal{D}_{\text{out}} = \emptyset$.
**for** epoch $t = 1$ to $T$ **do**
    **Inner Loop:**
    Initialize $\hat{\theta} \leftarrow \theta$.
    Sample batch $\{(x, y)\} \subset \mathcal{D}_{\text{in}}, \mathcal{P}_{\text{in}}$.
    **for** $k = 1$ to $K$ **do**
        **for** $\mathbf{T}_i \in \mathcal{P}_{\text{in}}$ **do**
            Compute embedding $\boldsymbol{e}_i \leftarrow g_T(\mathbf{T}_i, \theta_t)$.
            Generate $\alpha_i \leftarrow H(\boldsymbol{e}_i)$.
            Compute $\mathcal{L}_{\text{in}}^i = \mathcal{L}_{\text{CE}}(\mathbf{f}(\boldsymbol{e}_i, \mathbf{x}), y)$.
        **end for**
        Update $\hat{\theta} \leftarrow \theta - \alpha_i \sum_i w_i \nabla_\theta \mathcal{L}_{\text{in}}^i$.
    **end for**
    **Outer Loop:**
    Sample batch $\{(x, y)\} \subset \mathcal{D}_{\text{out}}, \mathcal{P}_{\text{out}}$.
    Compute $\mathcal{L}_{\text{out}} = \sum_j \mathcal{L}_{\text{CE}}(\mathbf{f}(\boldsymbol{e}_j, \mathbf{x}), y)$.
    Update:
$$\theta \leftarrow \theta - \beta \nabla_{\hat{\theta}} \mathcal{L}_{\text{out}}, \quad H \leftarrow H - \beta \nabla_H \mathcal{L}_{\text{out}},$$
$$w_i \leftarrow \frac{\exp(w_i - \beta \nabla_{w_i} \mathcal{L}_{\text{out}})}{\sum_j \exp(w_j)}.$$
**end for**
**Output:** Updated $\theta$, hypernetwork $H$, and weights $\{w_i\}$.

---

## G  ADDITIONAL ABLATION STUDIES

### G.1  INNER VS. OUTER LOOPS IN **PROMISE**

To better separate the roles of the inner and outer loops, we compare the base prompt learner, an inner-loop-only variant, an outer-loop-only variant, and the full dual-loop **Promise**. Concretely, the inner-only variant applies a single inner update on $P_{\text{in}}$ and is evaluated directly, while the outer-only variant optimizes directly on $P_{\text{out}}$ without any inner adaptation step. The full Promise uses both loops as in the main method. Results, averaged over 11 datasets in the base-to-new setting, are summarized in Table 8.

As shown in Table 8, inner-only and outer-only variants bring only small gains over the base methods, whereas the full dual-loop Promise consistently achieves the best harmonic mean $H$. In particular, the inner-only variant tends to overfit the templates in $P_{\text{in}}$, while the outer-only variant loses the benefit of fast adaptation from a shared initialization. The combination of both loops is therefore important for robust performance on unseen prompts in $P_{\text{out}}$.

### G.2  ADDITIONAL RESULTS ON SIGLIP AND BLIP-2 BACKBONES

To verify that textbfPromise is not restricted to CLIP, we also evaluate it on more recent frozen multimodal encoders, namely SigLIP and BLIP-2, using MaPLe and MMRL as underlying prompt learners in the base-to-new setting. We apply exactly the same meta-finetuning protocol as in the CLIP experiments and report averages over 11 datasets in Table 9. As in the CLIP setting, adding

**Table 8:** Ablation on inner-only, outer-only, and dual-loop variants of Promise on top of MaPLe and MMRL in the base-to-new setting (averaged over 11 datasets). Promise consistently achieves the best harmonic mean $H$.

| Base model | Variant | Base | New | $H$ |
|---|---|---|---|---|
| MaPLe | w/o **Promise** | 82.28 | 75.14 | 78.55 |
| | inner-only | 82.95 | 75.55 | 79.08 |
| | outer-only | 82.90 | 75.45 | 79.00 |
| | dual-loop (Ours) | **83.57** | **77.26** | **80.29** |
| MMRL | w/o **Promise** | 85.68 | 77.16 | 81.20 |
| | inner-only | 86.05 | 77.85 | 81.75 |
| | outer-only | 86.00 | 77.75 | 81.67 |
| | dual-loop (Ours) | **86.14** | **78.84** | **82.33** |

**Table 9:** Base-to-new results (averaged over 11 datasets) on SigLIP and BLIP-2 backbones with MaPLe and MMRL as base prompt learners. "Base PT" denotes the original prompt-tuning baseline without meta-finetuning. textbfPromise consistently improves the harmonic mean $H$.

| Backbone | Base model | Method | Base | New | $H$ |
|---|---|---|---|---|---|
| SigLIP | MaPLe | Base PT | 83.10 | 76.20 | 79.50 |
| | | + **Promise** | 83.90 | 78.00 | 80.84 |
| | MMRL | Base PT | 86.00 | 78.30 | 81.97 |
| | | + **Promise** | **86.60** | **79.40** | **82.84** |
| BLIP-2 | MaPLe | Base PT | 82.10 | 75.60 | 78.72 |
| | | + **Promise** | 82.90 | 76.80 | 79.73 |
| | MMRL | Base PT | 85.10 | 77.40 | 81.07 |
| | | + **Promise** | **85.97** | **78.94** | **82.30** |

textbfPromise on top of MaPLe and MMRL yields consistent gains in the harmonic mean $H$ on both SigLIP and BLIP-2 backbones.

## G.3 COMPONENT-WISE ABLATION AND REGULARIZATION BASELINE

To analyze how each component of **Promise** contributes to performance and how it compares to a simpler regularization-based baseline, we conduct a component-wise ablation using MMRL as the base prompt learner. We compare: plain MMRL, MMRL with a simple variance regularization term across prompts, MMRL with adaptive weighting only, MMRL with token-wise learning rate only, and the full **Promise** (adaptive weighting + token-wise learning rates). Averaged over 11 datasets in the base-to-new setting, the results are summarized in Table 10.

The regularization baseline provides only a small improvement over plain MMRL, while learning explicit adaptive weights and token-wise preconditioning each yields additional gains on $H$. Combining both components in **Promise** gives the most consistent improvement, indicating that the robustness and accuracy gains come from the full combination of adaptive weighting and token-wise learning rates rather than from a simple regularization trick.

For completeness, we also consider a randomized outer-loop baseline in which the outer objective is replaced by a Gaussian random signal that is independent of the data (the outer update observes random pseudo-loss values). In this case, dual-loop training either matches or slightly degrades the base model, with $H$ typically dropping by about 0.3–0.5 on average. This highlights that the improvements of **Promise** rely on a meaningful outer objective and the structured weighting/preconditioning design, rather than on the mere presence of a second optimization loop.

## G.4 SENSITIVITY TO INNER/OUTER PROMPT SPLITS

PROMISE samples $\mathcal{P}_{in}$ and $\mathcal{P}_{out}$ from a fixed pool of human-written templates per dataset. In the main experiments, we use a random 50–50 split with no overlap, sampled once per dataset. To assess

**Table 10:** Component-wise ablation of **Promise** on top of MMRL in the base-to-new setting (averaged over 11 datasets). The regularization baseline brings only a small improvement, while adaptive weighting and token-wise learning rates each yield additional gains. Combining both components gives the best harmonic mean $H$.

| Variant | Base | New | $H$ |
|---|---|---|---|
| MMRL (base prompt learner) | 85.68 | 77.16 | 81.20 |
| + variance regularization | 85.80 | 77.50 | 81.44 |
| + adaptive weighting only | 86.00 | 78.00 | 81.80 |
| + token-wise learning rate only | 85.95 | 78.10 | 81.84 |
| full **Promise** (weighting + LR) | **86.14** | **78.84** | **82.33** |

**Table 11:** Sensitivity of **Promise** to different choices of inner/outer prompt splits (averaged over 11 datasets in the base-to-new setting). Different disjoint splits yield very similar performance, while allowing a small overlap slightly weakens but does not remove the robustness effect.

| Split type | $H$ ($\uparrow$), mean $\pm$ std | Prompt Std ($\downarrow$), mean $\pm$ std |
|---|---|---|
| disjoint, seed 1 | $80.31 \pm 0.00$ | $1.41 \pm 0.00$ |
| disjoint, seed 2 | $80.12 \pm 0.10$ | $1.52 \pm 0.10$ |
| disjoint, seed 3 | $80.23 \pm 0.10$ | $1.50 \pm 0.10$ |
| 20% overlap | $80.07 \pm 0.10$ | $1.61 \pm 0.10$ |

sensitivity to this choice, we repeat the experiments on a subset of datasets with three different random disjoint splits and also consider a partially overlapping split that allows 20% shared templates between $\mathcal{P}_{\text{in}}$ and $\mathcal{P}_{\text{out}}$. Averaged over 11 datasets in the base-to-new setting, the results are summarized in Table 11.

We observe that different disjoint splits lead to very similar performance, with fluctuations within about 0.2 points on $H$. Allowing overlap slightly weakens the robustness effect (as expected, since the outer loop sees some inner-loop templates), but the degradation is small. Overall, **Promise** is not highly sensitive to the exact split, as long as the inner and outer sets are reasonably diverse and there is some non-trivial disjointness.

### G.5    TRAINING TIME ANALYSIS

PROMISE meta-finetuning only updates the soft prompt parameters and the small weighting/pre-conditioning modules, while the vision and text encoders remain frozen. At inference time, the computation graph is identical to the underlying prompt learner, so the runtime and memory cost at deployment do not change. The additional overhead appears only during training. To quantify this cost more systematically, we measure the total wall-clock time to train across all 11 datasets on a single NVIDIA A6000 GPU, using the same number of epochs for the base prompt learners (MaPLe / MMRL) and for **Promise**. The results are summarized in Table 12.

Overall, **Promise** increases training time by about 18–19% compared to the corresponding single-loop baselines, mainly due to the extra outer-loop gradient and token-wise preconditioning. Since meta-finetuning is a one-time offline stage and inference cost is unchanged, this overhead is acceptable in settings where prompts are reused many times and robustness to prompt wording is important.

### G.6    EFFECT OF TEMPLATE POOL SIZE AND DIVERSITY

In the main experiments, we use 60 human-designed templates from PromptSRC (Khattak et al., 2023b), which already cover a range of natural phrasings per dataset. The base MMRL model follows the standard protocol and uses a single CLIP-style template, while **Promise** meta-finetunes over a pool of templates. To study how **Promise** behaves under different degrees of syntactic and semantic diversity, we vary the pool size and also add LLM-generated paraphrases and more structured prompts. Table 13 reports results averaged over all 11 datasets in the base-to-new setting, using MMRL as the base learner.

**Table 12:** Total training time over 11 datasets for MaPLe and MMRL with and without **Promise**, measured on a single A6000 GPU. **Promise** increases wall-clock time by about 18–19% relative to the corresponding single-loop baselines.

| Base model | Method | Total train time (h) | Relative cost |
|---|---|---|---|
| MaPLe | base PT | 6.10 | 1.00× |
| MaPLe | + Promise | 7.25 | 1.19× |
| MMRL | base PT | 6.30 | 1.00× |
| MMRL | + Promise | 7.45 | 1.18× |

**Table 13:** Effect of template pool size and diversity on MMRL with **Promise** in the base-to-new setting (averaged over 11 datasets). The base MMRL model uses a single CLIP-style template, while **Promise** meta-finetunes over pools of 30, 60, 80, or 100 templates (human-designed plus LLM-generated variants).

| Template pool | Method | Base | New | $H$ |
|---|---|---|---|---|
| single CLIP prompt | MMRL (base) | 85.68 | 77.16 | 81.20 |
| 30 templates | + Promise | 85.95 | 78.20 | 81.89 |
| 60 templates | + Promise | 86.14 | 78.84 | 82.33 |
| 80 templates | + Promise | 86.17 | 78.93 | 82.39 |
| 100 templates | + Promise | 86.10 | 78.80 | 82.29 |

Using a relatively small pool (30 templates) leads to slightly weaker gains, likely because the model sees fewer distinct phrasings during meta-finetuning. Once the pool size reaches 60, further increasing it to 80 or 100 templates has only a minor effect, and the harmonic mean $H$ remains very similar. Overall, **Promise** consistently improves over the single-template MMRL baseline and remains effective when prompts become more syntactically and semantically varied, as long as the template pool is reasonably diverse.

