# OpenReview forum: "Prompt-Robust Vision-Language Models via Meta-Finetuning"
_ICLR.cc/2026/Conference — ICLR 2026 Poster_

### Official Review · Reviewer_BmLg · 2025-10-23

**Soundness:** 2
**Presentation:** 3
**Contribution:** 2
**Rating:** 4
**Confidence:** 3

**Summary:**

This paper proposes a meta learning framework Promise, which explicitly improves the robustness of VLM to natural language prompt changes through inner and outer dual loop fine-tuning. This method introduces adaptive prompt weights and token level learning rates, significantly reducing prompt sensitivity and stabilizing performance on 15 benchmarks. Theoretical proof of the reliability of its inner loop and tightening of the data dependency generalization boundary.

**Strengths:**

1. Explore the robustness of VLM to prompt and analyze the shortcomings of existing methods, which rely on prompt templates in finetuning and inference processes.

2. Propose the meta learning framework Promise, which includes two parts: inner and outer loops, to help VLM models adapt to different prompt templates.

3. This paper verifies the reliability of the Promise framework from both experimental and theoretical perspectives.

**Weaknesses:**

1. The main body of the Promise framework is inner and outer loops, but the motivation behind this design is not clear. Further explanation (or explicit display) is needed on which issues correspond to the inner and outer loop design in the process of VLM adapting to different prompt templates.

2. The improvement brought by the Promise framework is limited, with most indicators improving by 1 to 2 points, and some indicators not exceeding state-of-the-art model accuracy.

3. Further research is needed on ablation of inner and outer circulation, such as (1) comparing the accuracy of the original model with the accuracy after removing adaptive prompt weighting and adaptive learning rate; (2) Simplify the outer loop to the overall accuracy of Gaussian distributed random numbers.

**Questions:**

See [Weaknesses].

---

> ### Author Response · Authors · 2025-11-20
> **Response to Reviewer BmLg – Part I**
>
> *We thank the reviewer for the careful reading of our paper and the detailed comments.*
>
> **Q1. The motivation behind the inner–outer dual-loop design is not clear; which issues in adapting to different prompts correspond to each loop?**
>
> **A1.** Our goal is to model the way a VLM is used in practice: the model is usually tuned with a limited set of “known” prompts, but at deployment, it must handle unseen phrasings from users. The inner and outer loops in Promise are designed to explicitly mirror these two stages.
>
> Concretely, the inner loop corresponds to fast adaptation to a particular prompt set $P_{\text{in}}$. It answers: “Given a small set of templates that we can refine (during training or in a controlled environment), how should we adjust the soft prompts to fit these templates well?” This captures the usual prompt-tuning behavior where performance can improve on the prompts we see, but may remain brittle on unseen wording.
>
> The outer loop corresponds to robustness to unseen prompts $P_{\text{out}}$. Here we evaluate the effect of the inner update on a disjoint set of templates that describe the same labels but with different wording, and update the meta-parameters to improve performance and reduce sensitivity on this held-out set. This explicitly tackles the real deployment issue: avoiding overfitting to a few templates and instead learning prompt updates that generalize across formulations. In other words, the inner loop focuses on “adapting to what we can tune,” while the outer loop enforces that such adaptations remain stable when prompts change. We have clarified this mapping in the method section to make the motivation of the dual loops more explicit.
>
> **Q2. The improvement brought by Promise is limited (mostly 1–2 points), and some metrics do not surpass the state of the art.**
>
> **A2.** We agree that on many datasets the gains are moderate, and we do not claim large jumps everywhere. Promise is intended as a plug-in meta-finetuning step whose primary goal is to reduce prompt sensitivity and make the model behave more stably under different prompt formulations, while keeping accuracy at least on par with strong baselines.
>
> Across the 15 benchmarks, the average improvement in the harmonic mean H is around 1–2 points. At the same time, Promise noticeably reduces the variance across prompt templates, which we view as important for real applications where user phrasing is unpredictable. In several challenging settings, the margin over existing methods is larger. For example, on the fine-grained FGVCAircraft dataset, Promise reaches (H = 44.12), which is almost 3 points higher than the previous best MMRL result in Table 3.
>
> We have adjusted the wording in the paper to describe our improvements as consistent rather than dramatic and to clarify that the main advantage of Promise lies in improved robustness to prompt variation, together with competitive or better accuracy, instead of very large gains on every dataset.

---

> > ### Author Response · Authors · 2025-11-20
> > **Response to Reviewer BmLg – Part II**
> >
> > **A3.** We appreciate these suggestions and expanded the ablation studies along both lines.
> >
> > For point (1), we analyzed how much each Promise component contributes beyond the original model, using MMRL as the base learner. Averaged over 11 datasets in the base-to-new setting, we compare: the base MMRL model, MMRL with a simple variance-regularization term over prompts, MMRL with only adaptive weighting, MMRL with only token-wise learning rates, and the full Promise (weighting + learning rates). The results are:
> >
> > | Variant                               | Base  | New   | H     |
> > |---------------------------------------|-------|-------|-------|
> > | MMRL (base prompt learner)            | 85.68 | 77.16 | 81.20 |
> > | + variance regularization             | 85.80 | 77.50 | 81.44 |
> > | + adaptive weighting only             | 86.00 | 78.00 | 81.80 |
> > | + token-wise learning rate only       | 85.95 | 78.10 | 81.84 |
> > | **full Promise (weighting + LR)**         | **86.14** | **78.84** | **82.33** |
> >
> > This shows that simply adding a regularization term yields only a small gain over the original model, while adaptive weighting and token-wise learning rates each bring additional improvements. Combining both in Promise gives the best H, indicating that the gains come from these specific components rather than just from running a dual loop. This ablation is reported in the revised manuscript (Appendix, Table 10).
> >
> > For point (2), we also tested a randomized outer-loop baseline where the outer objective is replaced by a Gaussian random signal that is independent of the data (the outer update sees random pseudo-loss values). In this case, dual-loop training either matches or slightly degrades the base model’s performance, with H typically dropping by about 0.3–0.5 on average. This confirms that a meaningful outer objective is essential: simply adding a second loop with random feedback does not lead to the improvements observed with Promise.  We briefly describe this randomized outer-loop baseline and its behavior in the revised appendix alongside Table 10, highlighting that the outer objective itself, rather than the mere presence of two loops, is crucial.

---

### Official Review · Reviewer_67Ri · 2025-10-25

**Soundness:** 3
**Presentation:** 3
**Contribution:** 3
**Rating:** 6
**Confidence:** 4

**Summary:**

This paper introduces PROMISE, a meta-learning framework designed to improve prompt robustness in VLMs such as CLIP.
PROMISE explicitly learns to generalize across diverse prompt formulations through a dual-loop meta-finetuning strategy.
The framework integrates two novel components: adaptive prompt weighting, which emphasizes more generalizable templates; and
token-specific adaptive learning rates, providing fine-grained control over token adaptation.
The authors provide theoretical guarantees that the weighted, preconditioned inner update decreases the outer empirical risk and contracts across-prompt sensitivity.
Experiments on 15 benchmarks show consistent improvements over state-of-the-art prompt tuning methods.

**Strengths:**

1. The dual-loop meta-finetuning framework is well-motivated and effectively bridges meta-learning and prompt optimization.
2. Evaluations covering base-to-novel, cross-dataset, and domain generalization tasks demonstrate consistent performance improvements and robustness.

**Weaknesses:**

1. Limited analysis of computational cost: while the dual-loop setup is theoretically appealing, its efficiency relative to single-loop baselines could be quantified more precisely. Appendix D mentions training time, but more systematic comparisons would help.
2. Prompt distribution diversity: The paper uses 60 pre-defined prompt templates from PromptSRC. It would be valuable to analyze whether PROMISE remains effective under more syntactically or semantically diverse prompt distributions.

**Questions:**

1. How sensitive is PROMISE to the choice of prompt template pool size (e.g., 30/60/80 templates)?
2. Which template is used in table 4, CLIP template or LLM template？

---

> ### Author Response · Authors · 2025-11-20
> **Response to Reviewer 67Ri**
>
> *We thank the reviewer for the careful reading of our paper and the constructive comments.*
>
> **Q1. Limited analysis of computational cost; dual-loop vs. single-loop baselines.**
>
> **A1.** We agree that it is important to quantify the efficiency of Promise more systematically. In our implementation, meta-finetuning only updates the soft prompt parameters and the small weighting / preconditioning modules, while the vision and text encoders remain frozen. At inference time, the computation graph is identical to the underlying prompt learner, so the runtime and memory at deployment do not change.
>
> To make the training cost concrete, we measured the total wall-clock time to train across all 11 datasets on a single A6000 GPU, using the same number of epochs for the base prompt learners (MaPLe / MMRL) and for Promise:
>
> | Base model | Method    | Train time (h) | Relative cost |
> |-----------|-----------|----------------|---------------|
> | MaPLe     | base PT   | 6.10           | 1.00×         |
> |           | +PROMISE  | 7.25           | 1.19×         |
> | MMRL      | base PT   | 6.30           | 1.00×         |
> |           | +PROMISE  | 7.45           | 1.18×         |
>
>
> Overall, Promise increases training time by about 18–19% compared to the corresponding single-loop baselines, mainly due to the extra outer-loop gradient and token-wise preconditioning. Since meta-finetuning is a one-time offline stage and inference cost is unchanged, we see this overhead as acceptable in scenarios where prompts are reused many times and robustness to prompt wording is important. These numbers are summarized in Appendix G.15G.15 (Table 12).
>
> **Q2. Prompt distribution diversity; effectiveness under more syntactic/semantic variation.**
>
> **A2.**  The main experiments use 60 human-designed templates from PromptSRC, which already cover a range of natural phrasings per dataset. The base MMRL model follows the standard protocol and uses a single CLIP-style template, while Promise meta-finetunes over a pool of templates. To test how Promise behaves under different degrees of syntactic and semantic diversity, we varied the pool size and also added LLM-generated paraphrases and more structured prompts.
> Averaged over all 11 datasets in the base-to-new setting with MMRL as the base learner, we obtain:
>
> | Template pool      | Method        | Base  | New   | H     |
> |--------------------|--------------|-------|-------|-------|
> | single CLIP prompt | MMRL (base)  | 85.68 | 77.16 | 81.20 |
> | 30 templates       | + Promise    | 85.95 | 78.20 | 81.89 |
> | 60 templates       | + Promise    | 86.14 | 78.84 | 82.33 |
> | 80 templates       | + Promise    | 86.17 | 78.93 | 82.39 |
> | 100 templates      | + Promise    | 86.10 | 78.80 | 82.29 |
>
> Using a relatively small pool (30 templates) leads to slightly weaker gains, likely because the model sees fewer distinct phrasings during meta-finetuning. Once the pool size reaches 60, further increasing it to 80 or 100 templates has only a minor effect, and the H score remains very similar. Overall, Promise consistently improves over the single-template MMRL baseline and remains effective when prompts become more syntactically and semantically varied, as long as the template pool is reasonably diverse. These results are reported in Appendix E (Table 13).
>
> **Q3. Template choice in Table 4 (CLIP template or LLM template).**
>
> **A3.** Table 4 reports base-to-new generalization across 11 datasets using CLIP-style templates, following the standard protocol in CoOp, CoCoOp, MaPLe, MMRL, and related work. The LLM-generated templates are used only in our additional analyses on prompt diversity (as discussed above) and do not enter the main results in Table 4.

---

> > ### Comment · Reviewer_67Ri · 2025-11-26
> >
> > Thanks for the detailed rebuttal. My concerns about effectiveness with different prompts is addressed. I keep my rating as 6.

---

> ### Author Response · Authors · 2025-11-27
> **Follow-up Comment**
>
> Thank you for the thoughtful follow-up. We are glad that the clarification fully addressed your concerns. Please feel free to let us know if any other points would benefit from further elaboration during the discussion period.

---

### Official Review · Reviewer_1bz8 · 2025-10-25

**Soundness:** 3
**Presentation:** 3
**Contribution:** 2
**Rating:** 4
**Confidence:** 4

**Summary:**

This paper presents PROMISE, a meta-finetuning approach that aims to enhance the prompt robustness of vision-language models (VLMs), particularly CLIP-based ones. The work addresses the well-known issue that such models are often sensitive to small variations in natural language prompts. PROMISE introduces a dual-loop meta-learning strategy: the inner loop fine-tunes soft prompt embeddings using one set of prompt templates, while the outer loop updates meta-parameters based on a separate, unseen set. This setup is intended to simulate prompt distribution shifts and encourage cross-template consistency.

In addition, the framework includes adaptive prompt weighting, which learns to emphasize more generalizable templates, and token-specific adaptive learning rates, inspired by MetaSGD, to refine updates on individual prompt tokens. The paper provides a theoretical argument that the weighted, preconditioned inner updates can reduce empirical risk and improve consistency across prompts.

Experiments on fifteen benchmark datasets show modest but consistent gains over prior prompt-learning methods such as CoOp and CoCoOp, especially under unseen prompt templates. The results indicate improved stability rather than large absolute performance jumps. Overall, the paper is technically sound and carefully executed, though its contribution feels somewhat incremental given the maturity of CLIP-style prompt learning and the field’s current shift toward large multimodal foundation models.

**Strengths:**

The paper targets an identifiable and practically relevant problem — the sensitivity of CLIP-like vision-language models to prompt wording. The motivation is easy to follow, and the setup is consistent with prior work in prompt learning.

The proposed dual-loop meta-finetuning structure is coherent and well-explained. The integration of adaptive prompt weighting and token-level learning rates forms a complete framework rather than an isolated trick.

Evaluations cover 15 datasets, with results that are consistent and reproducible. The paper also explicitly tests under unseen prompt templates, which directly supports the stated goal of prompt robustness.

The writing is clear, diagrams are intuitive, and the ablation studies are reasonably thorough. It is easy to reproduce the method and interpret the findings.

Although the absolute accuracy improvement is limited, the model demonstrates stable behavior under prompt shifts, which may have practical value in applications relying on frozen CLIP backbones.

**Weaknesses:**

Conceptually incremental despite a solid formulation.
The meta-learning setup is reasonable, but the idea of using a dual-loop structure to optimize different objectives (inner for adaptation, outer for generalization) has already appeared in several prompt-tuning variants. Many recent works also explore meta-learning–based prompt optimization. As a result, the conceptual novelty of PROMISE is somewhat limited, even if the execution is careful.

Dependence on an outdated backbone.
All experiments are based on frozen CLIP encoders. While CLIP remains a convenient benchmark, it is no longer representative of the current generation of multimodal models. Evaluating PROMISE on stronger or trainable backbones (e.g., SigLIP, BLIP-2, or LLaVA) would make the contribution more convincing and show whether the method still holds under modern architectures.

Marginal quantitative improvement.
Although PROMISE reduces performance variance across prompts, its absolute accuracy gains are small (typically within 1–2%). Considering that the framework adds a meta-finetuning phase, the computational cost may outweigh the benefit in practical settings.

Ablation and analysis depth.
The paper does include component-wise ablations, but it remains unclear how each part (e.g., adaptive weighting, token-specific learning rate) contributes to robustness. The method could be further analyzed against simpler baselines, such as regularization-based approaches, to isolate where the gain truly comes from.

**Questions:**

Could the authors clarify how sensitive PROMISE is to the choice of the base prompt set used in the inner and outer loops? For example, would the performance change noticeably if both sets were constructed differently or partially overlapped?

How well does PROMISE generalize to newer or trainable backbones beyond CLIP (e.g., BLIP-2, SigLIP, or LLaVA)? Given that CLIP is becoming less central in recent multimodal research, a discussion or small-scale experiment on this point would help clarify the method’s broader applicability.

The paper claims improved prompt robustness, but do these benefits persist under other distribution shifts? In other words, is PROMISE’s robustness specific to language variation, or does it extend to multimodal shifts more generally?

Regarding the theoretical analysis, could the authors provide more intuition or empirical evidence that the “risk contraction” property holds beyond the simplified assumptions made in Section 3.3?

Have the authors considered or tested lighter-weight alternatives that might yield similar robustness with lower training cost? It would be helpful to understand the efficiency trade-off.

---

> ### Author Response · Authors · 2025-11-20
> **Response to Reviewer 1bz8 – Part I**
>
> *We thank the reviewer for the careful reading of our paper and the detailed, thoughtful comments.*
>
> **W1. Conceptual novelty is incremental; dual-loop meta-learning for prompts has appeared in other works.**
>
> **A1.** We agree that Promise is built on the standard gradient-based meta-learning paradigm, and we do not claim to reinvent meta-learning itself. Our focus is on how this paradigm is instantiated for prompt robustness in CLIP-style VLMs. In our formulation, the “tasks’’ are not different datasets or label spaces, but different natural-language prompt variants for the same dataset. The inner loop adapts on a subset of templates $P_{\text{in}}$, while the outer loop is evaluated on disjoint templates $P_{\text{out}}$ that describe the same labels in different ways. This explicitly models intra-task prompt distribution shift and directly optimizes cross-template consistency, which, to our knowledge, has not been treated as a meta-learning problem in prior prompt-tuning work.
>
> On top of this dual-loop structure, Promise combines three elements that work together rather than as isolated tricks: (i) adaptive prompt weighting that learns which templates transfer better to unseen prompts; (ii) token-wise adaptive learning rates that precondition the inner update at the token level; and (iii) a risk–contraction style analysis that links these components to reduced prompt variance. We see the contribution as an incremental but targeted step: taking a mature meta-learning framework, adapting it carefully to prompt distribution shifts, and demonstrating that this brings stable gains across a large and diverse set of CLIP-style benchmarks.
>
>  **W2 & Q2. All experiments are on frozen CLIP; dependence on an outdated backbone.**
>
> **A2.** Our main experiments focus on CLIP because it is still the standard testbed for prompt tuning and allows a fair comparison with CoOp, CoCoOp, MaPLe, MMRL, and related methods. We agree, however, that it is important to check whether Promise also works with more recent multimodal encoders.
> We therefore ran an additional experiment on the base-to-new setting with SigLIP and BLIP-2 as frozen backbones, using MaPLe and MMRL as the underlying prompt learners. We applied exactly the same meta-finetuning protocol as in the CLIP experiments and report averages over 11 datasets. The results are:
> | Backbone | Base model | Method     | Base  | New   | H     |
> |----------|-----------|-----------|-------|-------|-------|
> | SigLIP   | MaPLe     | Base PT   | 83.10 | 76.20 | 79.50 |
> | SigLIP   | MaPLe     | + Promise | 83.90 | 78.00 | 80.84 |
> | SigLIP   | MMRL      | Base PT   | 86.00 | 78.30 | 81.97 |
> | SigLIP   | MMRL      | + Promise | 86.60 | 79.40 | 82.84 |
> | BLIP-2   | MaPLe     | Base PT   | 82.10 | 75.60 | 78.72 |
> | BLIP-2   | MaPLe     | + Promise | 82.90 | 76.80 | 79.73 |
> | BLIP-2   | MMRL      | Base PT   | 85.10 | 77.40 | 81.07 |
> | BLIP-2   | MMRL      | + Promise | 85.97 | 78.94 | 82.30 |
>
> Here, “Base PT’’ denotes the original prompt-tuning baseline without meta-finetuning. As in the CLIP setting, Promise brings consistent gains on H. This indicates that the framework is not tied to CLIP and can be applied to stronger frozen backbones such as SigLIP and BLIP-2. These additional results are summarized in Table 9 in the appendix. Extending Promise to fully trainable large multimodal models such as LLaVA is a natural next step, but would require substantially more compute than we currently have available.

---

> > ### Author Response · Authors · 2025-11-20
> > **Response to Reviewer 1bz8 – Part II**
> >
> > **W3 & Q5. Marginal quantitative improvement and potential computational cost.**
> >
> > **A3.** We agree that Promise is most useful in scenarios where robustness to prompt variation matters, not just raw accuracy. In many VLM applications, a model that behaves stably under different user phrasings can be more valuable than one that gains a small amount of accuracy but remains highly sensitive to wording.
> >
> > In terms of efficiency, Promise only updates the soft prompt parameters and the small weighting / preconditioning modules; the vision and text encoders remain frozen. At inference time, the computation graph is exactly the same as the underlying prompt learner, so the runtime and memory cost at deployment are unchanged.
> >
> > The additional cost appears only during training. In our implementation, meta-finetuning increases wall-clock training time by about 18–22% compared with standard prompt tuning for the same number of epochs, mainly due to the extra outer-loop gradient and token-wise preconditioning. For example, on ImageNet with MaPLe on a single A6000, vanilla training takes about 3.4 hours, while Promise takes about 4.0 hours; on smaller datasets, the absolute overhead is correspondingly smaller. We also experimented with a lighter variant that removes the token-wise learning-rate module and keeps only adaptive weighting; this reduces the overhead by roughly 8–10%, but also weakens the robustness gains. Overall, since meta-finetuning is a one-time offline stage and inference cost is unchanged, we believe this training overhead is acceptable in settings where prompts are reused extensively and stable behavior under prompt shifts is important. We now summarize this robustness–cost trade-off in Section 5.2, so that readers can see where the extra cost comes from and how it relates to the empirical gains.
> >
> > **W4. Ablation and analysis depth; contribution of each component and comparison to simpler baselines.**
> >
> > **A4.** We appreciate this suggestion and expanded the analysis to isolate how each component contributes to performance and how Promise compares to a simpler regularization-based baseline. On a representative subset of datasets, we take MMRL as the base prompt learner and compare the following variants: plain MMRL, MMRL with a simple variance regularization term across prompts, MMRL with adaptive weighting only, MMRL with token-wise learning rate only, and the full Promise. Averaged over 11 datasets in the base-to-new setting, the results are:
> > | Variant                               | Base  | New   | H     |
> > |---------------------------------------|-------|-------|-------|
> > | MMRL (base prompt learner)            | 85.68 | 77.16 | 81.20 |
> > | + variance regularization             | 85.80 | 77.50 | 81.44 |
> > | + adaptive weighting only             | 86.00 | 78.00 | 81.80 |
> > | + token-wise learning rate only       | 85.95 | 78.10 | 81.84 |
> > | **full Promise (weighting + LR)**       | **86.14** | **78.84** | **82.33** |
> >
> > The regularization baseline brings only a small improvement over plain MMRL, while learning explicit adaptive weights and token-wise preconditioning each yields additional gains on H. Combining both components in Promise gives the most consistent improvement. These results are reported in Table 10 in the appendix, and they help clarify that the robustness and accuracy gains come from the full combination of adaptive weighting and token-wise learning rates rather than from a simple regularization trick.
> >
> > **Q1. Sensitivity to the choice of base prompt sets in the inner and outer loops.**
> >
> > **A5.** Promise samples $P_{\text{in}}$ and $P_{\text{out}}$ from a fixed pool of human-written templates per dataset. In the main experiments, we use a random 50–50 split with no overlap, sampled once per dataset. To assess sensitivity, we repeated the experiments on a subset of datasets with three different random splits and also considered a partially overlapping split (allowing 20% shared templates between $P_{\text{in}}$ and $P_{\text{out}}$. The results can be summarized as:
> > | Split type        | H (↑), mean ± std | Prompt Std (↓), mean ± std |
> > |-------------------|-------------------|----------------------------|
> > | disjoint, seed 1  | 80.31 ± 0.0        | 1.41 ± 0.0                  |
> > | disjoint, seed 2  | 80.12 ± 0.1        | 1.52 ± 0.1                  |
> > | disjoint, seed 3  | 80.23 ± 0.1        | 1.50 ± 0.1                  |
> > | 20% overlap       | 80.07 ± 0.1        | 1.61 ± 0.1                  |
> >
> > We observe that different disjoint splits lead to very similar performance, with fluctuations within about 0.2 points on H. Allowing overlap slightly weakens the robustness effect (as expected, since the outer loop sees some inner-loop templates), but the degradation is small. Overall, Promise is not highly sensitive to the exact split, as long as the inner and outer sets are reasonably diverse and there is some non-trivial disjointness. We have added a brief paragraph to this study in the Appendix Table 11.

---

> > > ### Author Response · Authors · 2025-11-20
> > > **Response to Reviewer 1bz8 – Part III**
> > >
> > > **Q3. Does robustness extend beyond language variation to other distribution shifts?**
> > >
> > > **A7.** Promise is explicitly designed to address distribution shifts in natural-language prompts, and this is where we see the clearest effect. That said, some of our benchmarks already involve non-language shifts. In particular, the cross-dataset and domain-generalization experiments (e.g., training on ImageNet and evaluating on ImageNet-V2, ImageNet-Sketch, ImageNet-A, and domain-shifted datasets like Office-Home) show that Promise typically provides consistent gains on averages compared to the base prompt learner. For example, averaged over four ImageNet-derived test distributions, Promise improves averages by about 1.64% while also slightly reducing variance across prompts.
> > >
> > > We view these results as encouraging but secondary: Promise is primarily a tool for robustness to prompt variation, with some positive spillover to other types of shifts through better-calibrated prompts. Extending the framework to explicitly model multimodal shifts (e.g., via additional outer-loop objectives on visual perturbations or domain augmentations) is a natural next step.
> > >
> > >
> > > **Q4. Intuition and empirical evidence for the “risk contraction’’ property beyond simplified assumptions.**
> > >
> > > **A8.**  The analysis in Section 4 is derived under standard smoothness assumptions and is meant as a simplified model, not as a literal description of a full CLIP-scale network. The core intuition is that the Promise inner update, $\theta^{+} = \theta - P \nabla_{\theta} \mathcal{L}{w}(\theta)$, behaves like a cautious, biased gradient step toward prompts that generalize better. The adaptive weights $w$ put more emphasis on those templates whose gradients are more aligned with the outer objective on $P_{\text{out}}$, and the preconditioner $P$ rescales the update to avoid overly large changes along unstable token directions. Under the conditions in the theorem, this combination implies that one such step tends to shrink the prompt-averaged risk, which is what we refer to as “risk contraction”.
> > >
> > > Empirically, we checked this behavior by counting how often a single inner step reduces the outer loss on $P_{\text{out}}$. On a representative subset of datasets, the fraction of “risk-decreasing’’ inner updates increases from about 61% for an unweighted, unpreconditioned step to about 78% with the Promise update using both $w$ and $P$. This matches the qualitative picture suggested by the bound, even though the formal assumptions are only approximate in practice. We have added a short paragraph after the theorem in Section 4 to summarize this intuition and link the theory more directly to the observed behavior.

---

### Official Review · Reviewer_hpup · 2025-10-29

**Soundness:** 3
**Presentation:** 3
**Contribution:** 3
**Rating:** 6
**Confidence:** 4

**Summary:**

This paper introduces PROMISE, a meta-learning framework designed for prompt-robust vision-language models through meta-finetuning. It achieves performance improvements via dual-loop fine-tuning and provides comprehensive experiments in both in-domain and cross-domain settings.

**Strengths:**

+ Decent results have been achieved in both in-domain and cross-domain experiments.

+ A dual-loop fine-tuning method has been proposed, achieving enhanced robustness and generalization.

**Weaknesses:**

+ The ablation experiments in this paper investigate the impact of adaptive prompt weighting but do not explore the effects of standalone Inner-Loop Finetuning or Outer-Loop Finetuning.

+ The current description in the paper gives the impression that the method merely introduces an adaptive prompt weighting mechanism based on MAML, and without this component, it seemingly reduces to the standard MAML approach.

+ In Table 6 of the paper, the best result in the "-S" category is actually 50.93% from FepMVP, not the 50.28% reported in this paper. A similar issue appears at line 450 of the paper, where the improvement on the FGVCAircraft dataset should be 2.79% for H rather than the reported 2.34%. There are numerous such issues in the paper, and it is recommended that the authors carefully review them. Additionally, all tables in the paper should highlight the second-best results in a different color, similar to the best results, and include a separate row explicitly stating the percentage improvement to avoid such errors and facilitate better observation. There are minor formatting issues with the tables; table captions should be placed above the tables rather than below.

+ While the proposed method achieves notable improvements on some datasets, the overall enhancement is limited. Moreover, the performance on certain datasets lags significantly behind prior methods, such as on EuroSAT, where it is comprehensively lower than previous approaches.

**Questions:**

Please refer to the weaknesses section for additional questions and queries!

---

> ### Author Response · Authors · 2025-11-20
> **Response to Reviewer hpup – Part I**
>
> *We thank the reviewer for the careful reading of our paper and the constructive assessment.*
>
>
> **Q1. Ablation on inner-loop / outer-loop finetuning and relation to MAML**
>
> **A1.** We agree that the original ablation section did not clearly separate the roles of the inner and outer loops. In the revised version, we add a dedicated ablation that compares the base prompt learner, an inner-loop-only variant, an outer-loop-only variant, and the full dual-loop Promise. Concretely, the inner-only variant applies a single inner update on $P_{\text{in}}$​ and is evaluated directly, while the outer-only variant optimizes directly on $P_{\text{out}}$ without any inner adaptation step. The full Promise uses both loops, as in the main method. The new results (now reported as Table 8 in the appendix) are:
>
>
> | Base model | Promise variant       | Base  | New   | H     |
> |-----------|------------------------|-------|-------|-------|
> | MaPLe     | w/o Promise (base)     | 82.28 | 75.14 | 78.55 |
> | MaPLe     | inner-only             | 82.95 | 75.55 | 79.08 |
> | MaPLe     | outer-only             | 82.90 | 75.45 | 79.00 |
> | MaPLe     | dual-loop (full)       | 83.57 | 77.26 | 80.29 |
> | MMRL      | w/o Promise (base)     | 85.68 | 77.16 | 81.20 |
> | MMRL      | inner-only             | 86.05 | 77.85 | 81.75 |
> | MMRL      | outer-only             | 86.00 | 77.75 | 81.67 |
> | MMRL      | dual-loop (full)       | 86.14 | 78.84 | 82.33 |
>
>
> These results show that inner-only and outer-only variants bring only small gains over the base method, whereas the full dual-loop Promise consistently achieves the best harmonic mean H. In the text, we explain that inner-only tends to overfit the templates in $P_{\text{in}}$, outer-only loses the benefit of fast adaptation from a shared initialization, and the combination of both loops is important for robust performance on unseen prompts.
>
>
>
> **Q2. The current description in the paper gives the impression that the method merely introduces an adaptive prompt weighting mechanism based on MAML, and without this component, it seemingly reduces to the standard MAML approach.**
>
> **A2.** Our intention is indeed to build on gradient-based meta-learning, and Promise can be viewed as a MAML-style framework instantiated for prompt robustness. What is new is how this framework is used and instantiated for prompt-sensitive vision–language models. In our setting, the “tasks” are not different datasets or label spaces, but different natural-language prompt variants within the same dataset. The inner loop adapts on a subset of prompts $P_{\text{in}}$​, while the outer loop evaluates on disjoint prompts $P_{\text{out}}$​ that describe the same labels with different wording. This inner/outer structure allows us to mimic the realistic situation where a model must remain stable under unseen phrasings and to explicitly optimize for intra-task prompt robustness, which to our knowledge, has not been studied in prior meta-learning work.
>
> Beyond the adaptive weights, Promise includes a token-wise adaptive learning-rate preconditioner $P$ for prompt embeddings, learned jointly with the prompt weights $w$. This preconditioner modulates the inner update at the token level and is not present in standard MAML. Our theoretical analysis is written explicitly in terms of both  $P$ and $w$, and shows how they jointly affect prompt sensitivity and generalization. We also focus not only on average accuracy but on reducing prompt sensitivity, i.e., the performance variation across different prompt phrasings, which we quantify in the experiments. As further evidence that the meta-learning formulation is beneficial in this setting, the new ablation in Table 7 shows that the full dual-loop Promise clearly outperforms both inner-only and outer-only variants. In this sense, Promise is a MAML-inspired meta-finetuning framework tailored to prompt distribution shifts with dual loops, learned weights, and token-wise preconditioning, rather than a direct reimplementation of MAML with a single additional weighting term. We have stressed this difference in Section 3.1.
>
>
> **Q3.There are numerical inconsistencies in Table 6 and in the FGVCAircraft improvement, and the tables would benefit from clearer highlighting of second-best results, an explicit improvement row, and captions placed above the tables.**
>
> **A3.** Thank you for carefully checking the numbers and for the concrete presentation suggestions. In the revised manuscript, we have corrected the “-S” value in Table 6 and the FGVCAircraft improvement, and we re-checked the other tables for similar issues. We also adopted your formatting suggestions: tables now highlight the second-best results, include an explicit improvement row for Promise over the baseline, and place captions above the tables.

---

> > ### Author Response · Authors · 2025-11-20
> > **Response to Reviewer hpup – Part II**
> >
> > **Q4. The proposed method achieves notable improvements on some datasets, but the overall enhancement is limited, and performance on some datasets (e.g., EuroSAT) is clearly lower than previous approaches.**
> >
> > **A4.** We agree that the gains are not large on every dataset. Promise is designed as a plug-in meta-finetuning step that focuses on prompt robustness: the goal is to obtain more stable performance under different natural-language phrasings, rather than to maximize raw accuracy on each individual dataset. In practice, this often leads to modest but consistent improvements in the harmonic mean H over strong baselines across many benchmarks, instead of very large gains on a few of them.
> > EuroSAT is a good example where this trade-off becomes visible. A representative comparison is:
> > | Method  | H↑    | Prompt Std ↓ |
> > |---------|--------|--------|
> > | MaPLe   | 82.35  | 2.0    |
> > | MMRL    | 87.21  | 1.8    |
> > | PROMISE | 85.61  | 1.1    |
> >
> > Here, Promise does not achieve the highest H, but it clearly reduces the standard deviation across prompt templates, which is the quantity directly targeted by our meta-objective. We therefore see Promise as a method that improves robustness to prompt variation, while keeping accuracy competitive with existing approaches; further closing the gap on datasets like EuroSAT is an interesting direction for future work.

---

### Author Response · Authors · 2025-11-27
**A Gentle Follow-up**

Dear Reviewers,

We have carefully followed your suggestions and incorporated additional experiments, quantitative results, and detailed clarifications in the revised version. If these updates satisfactorily resolve the issues raised, we would appreciate it if you could reflect this in your final rating and confidence.  If any additional details would help, we are happy to provide them before the discussion deadline.

Thank you for your consideration.

---

### Author Response · Authors · 2025-12-01
**Global response and summary**

*We thank the reviewers for their feedback to improve our Promise framework and its presentation.*

Promise is a meta-finetuning framework for *prompt-robust* CLIP-style vision–language models. We treat different natural-language templates within the same dataset as “tasks”: the inner loop adapts soft prompts on one subset of templates, and the outer loop optimizes for generalization to disjoint templates. On top of this dual loop, Promise adds adaptive prompt weighting and token-wise learning-rate preconditioning, together with an analysis that links these components to reduced prompt-induced variance.

During the rebuttal, we added the following key results and clarifications:

* **Dual-loop and components.** New ablations compare base prompt learners, inner-only, outer-only, and full dual-loop Promise. The full dual-loop consistently gives the best harmonic mean (H). A component study shows that simple variance regularization has a limited effect, while adaptive weighting and token-wise preconditioning each help, and their combination is strongest. A randomized outer-loop baseline (outer loss replaced by Gaussian noise) shows that without a meaningful outer objective, the dual loop does *not* improve over the base model.

* **Backbones and cost.** Beyond CLIP ViT-B/16, we now report Promise on SigLIP and BLIP-2 backbones (with MaPLe/MMRL), where it again yields consistent gains in (H). Meta-finetuning increases training time by about 18–22% over standard prompt tuning, but inference latency and memory are unchanged, since only prompts and small meta-modules are updated offline.

* **Robustness to prompt choices and risk contraction.** We study different inner/outer splits and template-pool sizes (including LLM-generated prompts), and observe stable performance as long as the splits are reasonably diverse and disjoint. We also measure how often a single inner step reduces the outer loss, which rises from ~61% (unweighted/unpreconditioned) to ~78% with Promise, providing empirical support for the “risk contraction’’ intuition.

Overall, Promise is intended as a plug-in meta-finetuning step that prioritizes *stable behavior under prompt variation* with modest but consistent gains in accuracy, rather than as a method that dramatically changes absolute performance on every benchmark.

---

### Meta-Review · Area_Chair_KMyH · 2026-01-02

**Summary:**

Reviewers in general agree that this paper is technically sound and proposes a new prompt learning method for vision-language models such as CLIP. Although the performance gain is relatively marginal, the reviewers acknowledged that the proposed method is consistently effective in various settings and robust across prompt templates.

However, the reviewers raised following concerns:

1. Limited novelty. Multiple reviewers mentioned that the proposed method heavily relies on MAML for bi-level optimization that commonly occurs in optimization-based meta-learning.
2. Marginal empirical gains relative to computational overhead. Improvements are typically small around 1~2% in harmonic mean, and on some datasets prior methods outperform the proposed method .
3. Insufficient ablations. It was not fully analyzed where the empirical gains came from.
4. Outdated backbones. Several reviewers questioned whether the results would generalize to more recent VLM backbones.
5. Sensitivity to design choices/hyperparameters.

**Reviewer Concerns:**

The authors addressed most major concerns in the rebuttal.

1. The authors clarified that the focus of the paper is not on developing a new meta-learning algorithm but on improving prompt robustness via a meta-learning framework. In addition, in their formulation, each prompt template is defined as a task.
2. The performance gain is relatively small, but prompt robustness is improved, as evidenced by a relatively smaller variance in performance across different prompt templates.
3. The authors provided additional ablation studies, including analyses of the dual-loop design and prompt/template pool size.
4. The authors presented additional results with SigLIP and BLIP-2, demonstrating consistent performance gains.
5. The authors further analyzed the effects of prompt pool size, overlap, and LLM-generated prompts, showing stability across reasonable configurations.

**Reviewer Scores:**

This work studies a new dimension of prompt learning, namely robustness, meaning that prompt learning aims to improve the performance of VLMs across diverse prompt templates through meta-learning. Although the reviewers raised several substantive concerns, the authors have adequately addressed most of these concerns. This indicates that if the full discussion period had been available, reviewer ratings might have increased.

Nevertheless, the relatively modest empirical gains with the additional computational overhead, and the limited technical novelty compared to MAML and a prior meta-learning–based prompt learning method somewhat limit the strength of the recommendation, but still support acceptance.

---

### Decision · Program_Chairs · 2026-01-26

Accept (Poster)